# Adipocytes control food intake and weight regain via Vacuolar-type H+ ATPase

Rizaldy C. Zapata[1], Maria Carretero[2], Felipe Castellani Gomes Reis [1],
Besma S. Chaudry [1], Jachelle Ofrecio[1], Dinghong Zhang [1], Roman Sasik[3],
Theodore Ciaraldi[1,4], Michael Petrascheck [2] & Olivia Osborn [1] ✉

Energy metabolism becomes dysregulated in individuals with obesity and many of these changes persist after weight loss and likely play a role in weight regain. In these studies, we use a mouse model of diet-induced obesity and weight loss to study the transcriptional memory of obesity. We found that the 'metabolic memory' of obesity is predominantly localized in adipocytes. Utilizing a *C. elegans*-based food intake assay, we identify 'metabolic memory' genes that play a role in food intake regulation. We show that expression of *ATP6v0a1*, a subunit of V-ATPase, is significantly induced in both obese mouse and human adipocytes that persists after weight loss. *C. elegans* mutants deficient in *Atp6v0A1/unc32* eat less than WT controls. Adipocyte-specific *Atp6v0a1* knockout mice have reduced food intake and gain less weight in response to HFD. Pharmacological disruption of V-ATPase assembly leads to decreased food intake and less weight re-gain. In summary, using a series of genetic tools from invertebrates to vertebrates, we identify ATP6v0a1 as a regulator of peripheral metabolic memory, providing a potential target for regulation of food intake, weight loss maintenance and the treatment of obesity.

Obesity is a major global health problem and predisposes individuals to many comorbidities that can reduce life expectancy[1]. While weight loss can be achieved through reduced caloric intake and/or increased physical activity, keeping the weight off is particularly challenging and many individuals regain weight[2]. For example, ~80% of individuals with obesity who lost 10% of their body weight will regain that lost weight within 1 year[3]. Weight loss results in physiological adaptations including perturbations in the levels of circulating appetite-related hormones[4], changes in energy expenditure[5] and increased appetite[6,7] that all contribute to weight regain[8]. Therefore, in the obese state, many signals controlling energy metabolism become dysregulated and further adaptive changes occur after weight loss that encourage weight regain to the higher body weight set point[9]. However, while these physiological adaptations have been well-described, the

molecular mechanisms underlying the drive for weight regain are poorly understood.

In our previous studies, we determined that obesity induces a lasting metabolic signature that does not resolve after weight loss and termed this concept as "metabolic memory"[10]. Numerous other studies has demonstrated that obesity triggers changes in the transcriptome[11–14], metabolome[10], proteome[15], and microbiome[16] that persist after weight loss. Notably, the adipose tissue plays an important role in the regulation of energy balance and has been the focus of many theories aimed at explaining the persistence of a biological drive to regain weight[10,13,17–19]. We found that the "metabolic memory of obesity" resides predominantly in adipose tissue where obesity-induced changes in metabolite levels persisted after weight loss, while in other metabolic tissues, including liver, muscle

[1]Division of Endocrinology and Metabolism, School of Medicine, University of California San Diego, La Jolla, CA 92093, USA. [2]Department of Molecular Medicine and Department of Neuroscience, The Scripps Research Institute, La Jolla, CA 92037, USA. [3]Center for Computational Biology & Bioinformatics, School of Medicine, University of California San Diego, La Jolla, CA 92093, USA. [4]VA San Diego Healthcare System, La Jolla, CA 92037, USA. ✉e-mail: oosborn@ucsd.edu

and hypothalamus, most of obesity-induced metabolite profiles return to pre-obese levels after weight loss[10]. The concept that adipose-derived metabolic memory plays a role in food intake and metabolism has been further described in a recent mouse study related to the benefits of caloric restriction[17]. This study found that switching mice from dietary restriction to *ad libitum* food intake acutely increases mortality, whereas the switch from *ad libitum* to dietary restriction causes only a weak and gradual increase in survival, suggesting a memory of earlier nutrition. Furthermore, RNA-seq profiling in key metabolic tissues demonstrated a largely refractory transcriptional and metabolic response in the white adipose tissue (WAT) in response to dietary restriction in mice that had previously been fed *ad libitum*[17].

Thus, we hypothesized that obesity-induced changes that persist after weight loss may play a role in feeding behavior and contribute to weight regain. In this study, we identify the first regulators of this peripheral metabolic memory through functional analysis of the transcriptional changes that occur in obese WAT that persist after weight loss and investigate the potential role of these metabolic memory genes in feeding behavior and weight regain.

## Results

### The "metabolic memory" of obesity resides primarily in adipocytes

To determine which obesity-induced genes are persistently dysregulated after weight loss, we conducted RNA-sequencing analysis of adipose tissue from lean (LFD fed) and obese (HFD fed) mice as well as formerly obese mice (HFD switched to LFD, 'SW') (Fig. 1a). To gain a deeper understanding of the specific cell type in which potential gene expression changes occur, we fractionated the WAT into adipocytes (Fig. 1b) and SVCs (immune cells) (Fig. 1c) and conducted RNA-seq analysis. HFD-induced obesity resulted in significant changes in gene expression compared with LFD fed mice in adipose tissue in both adipocytes (925, lfdr < 0.05) (Fig. 1b) and SVC cells (354, lfdr < 0.05) (Fig. 1c). Furthermore, after weight loss, many of the obesity-induced changes in the transcriptome were retained in both adipocytes (Fig. 1b) and SVCs (Fig. 1c). Specifically, in adipocytes 752/925 obesity-induced genes were persistently dysregulated genes after weight loss (Fig. 1b, genes listed in Supplemental Data 1) compared with 247/354 in SVCs (Fig. 1c, genes listed in Supplemental Data 2). Therefore, adipose tissue retains a metabolic memory of obesity and the majority of persistently dysregulated genes are localized in adipocytes (752), compared with SVCs (237).

### In vivo *C. elegans* based screen to identify metabolic memory genes with a role in food intake regulation

In this study, we chose to focus on mechanisms driving food intake as this plays a significant role in weight rebound in humans[6,7]. To identify metabolic memory genes that contribute to the regulation of food intake, we applied our high throughput food intake assay in *C. elegans*[20,21] (Fig. 1d). We first determined which of these 'metabolic memory genes', identified in mouse, have an orthologous gene in *C. elegans* (Supplemental Data File 3). We then further triaged this list by selecting genes with a high degree of conservation between the mammalian and worm ortholog using DRSC Integrative ortholog prediction tool (DIOPT)[22] (http://www.flyrnai.org/cgi-bin/DRSC_orthologs.pl) to rank the level of orthology (high rank = 202 genes, moderate rank = 254 genes, low rank = 101 genes and no worm ortholog = 195 genes, Supplemental Data 3). For 28 of these orthologous genes, *C. elegans* mutant strains were available and were screened to determine if loss of a candidate metabolic memory gene resulted in any changes in food intake. We identified 19 significantly hypophagic strains (ranging from 3–74% of normal food intake) and two hyperphagic strains (150–180% above normal

food intake) and eight strains with no effect on food intake compared to the control 'wild type' N2 strain (Fig. 1d, Supplemental Data 4). To identify mutant worm strains with lower food intake due to developmental defects or general sickness, we exposed each mutant to a strong hyperphagic signal (serotonin). The majority of hypophagic mutants, including *Atp6v01A/unc-32* mutant worms, were capable of a hyperphagic response after exposure to serotonin (15/19) (Fig. 1e, Supplemental Data 5). Only four of the 19 hypophagic mutants were unable to upregulate food intake when exposed to serotonin, suggesting they may have a developmental/sickness defects rendering them incapable of hyperphagia (Supplemental Data 5). This list of hypophagic mutants includes a wide range of genes with various functions including molecular binding, catalytic activity, transcriptional regulation and transport activity (Supplemental Data 6). *Atp6v01A/unc-32* codes for a lysosomal V-ATPase subunit gene and was of particular interest based on preliminary data from international mouse phenotyping consortium[23] showing male heterozygous *ATP6v0a1* whole body knockout mice (Atp6v0a1tm1b(EUCOMM)Hmgu) have significantly decreased fat mass ($p = 1.7 \times 10^{05}$) suggesting this gene may play an important role in body weight regulation.

### Adipose expression of ATP6v0a1 is elevated during obesity and persists after weight loss in both mouse and humans

V-ATPases are highly conserved in many species including *C. elegans*, mouse and human[24–26]. These proton pumps are composed of a cytosolic $V_1$ domain (subunits A-H) and a transmembrane $V_0$ domain (subunits a, c, c', c'', d, e) and serve to pump protons across membranes in several cellular organelles by hydrolyzing ATP. The expression of *Atp6v0a1* is specifically elevated in adipocytes of diet-induced obese mice and expression levels do not decrease after weight loss (Fig. 2a). The adipocyte expression levels of other subunits of the V-ATPase complex are also elevated in obesity and persist after weight loss (Fig. 2b). In addition, the expression of *Atp6v0a1* is highly elevated in multiple models of obesity including high-fat diet (HFD) fed mice (Fig. 2a) and leptin deficient (*ob/ob*) mice fed either normal chow or HFD (Fig. 2c). This confirms that obesity, whether induced by HFD, or leptin deficiency drives elevation in Atp6v0a1 (Fig. 2c).

We next determined whether the signature of metabolic memory genes identified in mouse were also elevated in adipocytes from patients with obesity compared with lean individuals[27]. Of the 752 metabolic memory genes identified in mouse, we found 141 genes with human orthologs and that were also significantly differentially expressed in the same direction in human adipocytes from patients with obesity (131 upregulated, 10 downregulated) (Supplemental Data 6). Importantly, gene (Fig. 2d) and protein (Fig. 2e) expression of ATP6v0a1 is also increased in adipocytes from participants with obesity compared with lean controls and expression persists after weight loss in obese adipose tissue (Fig. 2f).

While the regulation of V-ATPase can be achieved by changes in expression of V-ATPase subunit genes[28,29], activation and inactivation of the pump is controlled by the reversible association/dissociation of the V0 and V1 domains[24,30–33]. Various stimuli, including glucose stimulation[32], drive translocation of the Atp6V1 (V1) domain[34–36] to the organelle membrane, where is associates with the membrane-bound Atp6V0 (V0) domain resulting in increased V-ATPase activity and acidification of intracellular organelles[32]. Therefore, because the expression and location of the V1 and V0 domains is an important mechanism by which V-ATPase activity is controlled, we determined their abundance in the membrane and cytosolic fractions of WAT in lean, obese and formerly obese mice. Obesity resulted in a significant increase in the membrane and cytoplasmic abundance of both V0 and V1 domains (Fig. 2g, h) and the obesity-induced levels did not decrease after weight loss.

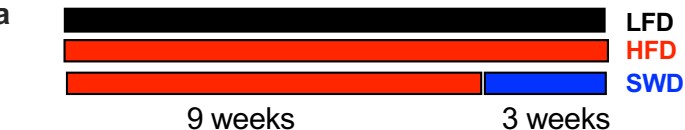

### Adipocyte-specific ablation of Atp6v0a1 decreased food intake and body weight

To study the role of adipocyte expression of *Atp6v0a1* in food intake, body weight and energy homeostasis, we generated adipocyte specific Atp6v0a1 KO mice. Adipocyte specific *Atp6v0a1* KO resulted in ~73% reduction in *Atp6V0a1* gene expression in the adipose tissue (Fig. 3a)

and 92% reduction at the protein level compared with WT mice (Fig. 3b). As expected, Atp6v0a1 levels in the SVCs of WAT(Fig. 3b) and whole brown fat (Fig. 3a) remain unchanged in KO compared with WT mice. KO mice fed normal chow diet had significantly lower average food intake (Fig. 3c) and weighed significantly less (Fig. 3d), with lower WAT as a percentage of total body weight (Fig. 3e), compared with WT

**Fig. 1 | Identification of metabolic memory genes in adipose tissue in formerly obese mice. a** Schematic diagram of diet switch mouse study. **b** RNA seq of adipocytes, **c** stromal vascular cells (SVCs), from lean(LF, black circles), obese (HF, red squares) and formerly obese (SW, blue triangles) mice. Gene expression was analyzed using the statistical algorithm limma[70] and differential expression between LF and HF fed mice determined as lfdr < 0.05, n = 3 per group. **d** *C. elegans* based food intake assay used to determine if genetic strains lacking orthologous mouse metabolic memory genes play a role in food intake. Strains with significantly different food intake from wild type "N2" strain was determined by one-way ANOVA

with Dunnetts multiple comparison test. Values are expressed as box and whisker plots, with boxes indicating the Q1 and Q3 ranges, center line representing the mean, and the whiskers as minimum and maximum values, *p < 0.05, n = 17–23 wells per strain. **e** *unc-32/ATP6v0a1* strain responds to hyperphagic stimulus serotonin as determined by unpaired student t-test (2 tailed), (average n = ~250 worms per strain, p < 0.0001). Values are expressed box and whisker plots, with boxes indicating the Q1 and Q3 ranges, center line representing the mean, and the whiskers as minimum and maximum values, *p < 0.05, n = 20–21 wells per strain. Statistical parameters can be found in the Supplementary Data 1–5.

control mice. There was no significant difference in food intake (Fig. 3c), body weight (Fig. 3d) or adipose tissue ATP6V0a1 gene and protein levels (Supplementary Information, Supplemental Fig. 1a, b) between Adiponectin-cre expressing mice and the Atp6v0a1^fl/fl (FF) mice.

### Adipocyte-specific ablation of Atp6v0a1 decreased food intake and body weight gain in HFD-fed mice

We then investigated the effect of adipocyte-specific Atp6v0a1 deletion on food intake and weight gain in response to HFD feeding. As an important control, we compared food intake and weight gain of FF and Adiponectin-cre mice when fed HFD (Fig. 3a, b) and found no differences between these two controls. Thus, in the remaining mouse experiments, we utilized the Adiponectin-cre mice as our 'WT' control. KO mice ate significantly less HFD (Fig. 4a) and had reduced body weight gain (Fig. 4b) compared with WT controls. In addition, fat mass (Fig. 4c) was lower and average adipocyte size was also smaller in the KO compared to WT (Fig. 4d). Respiratory quotient (Fig. 4f) and locomotor activity (Fig. 4g) were unchanged between KO and WT controls. Surprisingly, KO mice had slightly lower energy expenditure (Supplementary Information, Supplemental Fig. 2a, b), predominantly during the light period when mice are less active (Fig. 4e) compared with WT controls. Analysis of genes involved in BAT function (Supplementary Fig. 2d) were unchanged between KO and control mice. Furthermore, when BAT function is stimulated by either chronic exposure to cold[37,38], or administration of a beta-3 adrenergic agonist (CL-316,243)[38] the expression of ATP6v0a1 was unchanged. To further investigate whether the lower body weight in KOs is driven by lower food intake, we performed pair-feeding experiments. WT mice were given the same average quantity of HFD that KO mice normally consume (Supplementary Information, Supplemental Fig. 2e). After 7 days, pair fed-WT mice weighed significantly less that the WT *ad libitum* group, and had similar body weights as the KO group (Supplementary Information, Supplemental Fig. 3f). These data strongly suggest that the lower body weight observed in the KO mice is due to lower caloric intake and not due to potential differences in energy expenditure or changes in intestinal absorption.

Investigation of hypothalamic expression of appetite regulating genes by quantitative PCR revealed increased expression of satiety promoting pro-opiomelanocortin (*Pomc*) by approximately four-fold and a small but significant reduction in orexigenic gene neuropeptide Y (*Npy*) in the KO mice compared with WT control mice (Fig. 4h). In WAT (Fig. 4i), we observed increased adiponectin (*Adipoq*) and decreased leptin (*Lep*) expression in the KO compared with WT controls in a manner that tracked proportionally with body weight changes. Plasma levels of leptin, insulin, ghrelin and many other metabolic and inflammatory proteins were unchanged between groups (Supplementary Information, Supplemental Fig. 3a, b). Interestingly, we observed elevated levels of appetite suppressing hormone, growth differentiation factor-15 (Gdf15), in both WAT and the plasma, in the KO compared with WT controls (Fig. 4j). Recent literature has highlighted a mechanistic link between ER stress and Gdf-15 levels[39,40]. Notably, C/EBP homologous protein (Chop, Ddit3) has been shown to directly bind to the promoter of GDF15 and activates its transcription under ER stress conditions[39,40]. Therefore, to evaluate whether

Atp6V0a1 inhibition was associated with ER stress, we measured the expression of Ddit3 and activating transcription factor 4 (Atf4), which are classical markers of ER stress and found both genes to be upregulated in WAT from KO mice compared with controls (Fig. 4i).

HFD-induced obesity resulted in increase in the membrane and cytoplasmic abundance of both V0a1 and V1a domains (Fig. 2g, h) that persisted after weight loss. In the KO mice, the abundance of WAT Atp6v0a1 was significantly depleted in membrane fractions and also tended to be reduced in the cytoplasm (p = 0.12) of KO mice compared with WT controls (Fig. 4k).

### Adipocyte-specific ablation of Atp6v0a1 improved glucose metabolism in HFD-fed mice

HFD fed KO mice displayed significantly improved glucose tolerance (Fig. 5a), which is likely influenced by the lower body weight of the KO mice compared with controls (Fig. 5b). Basal and glucose stimulated insulin levels trended lower in the KO but were not statistically significant (Fig. 5c). Insulin sensitivity was significantly improved in KO compared with WT mice (Fig. 5d), with body weight also likely a key factor contributing to this improvement (Fig. 5e). To determine which tissues were primarily driving the improved insulin sensitivity, we analyzed phosphorylation of insulin signaling proteins in response to acute insulin stimulation. Acute insulin treatment resulted in significantly increased levels of phosphorylated Ser437 Akt specifically in the adipose tissue (Fig. 5f, g) while no significant change was observed in the liver or skeletal muscle of KO mice compared with WT mice (Fig. 5f, g).

### Bafilomycin treatment decreased food intake, weight gain and improved glucose tolerance

To determine if pharmacological inhibition of ATP6v0a1 effects food intake, weight gain and glucose homeostasis, we treated mice with bafilomycin (BFM). BFM is a macrolide antibiotic that functionally inhibits the activity of V-ATPases by preventing the association of the cytoplasmic and transmembrane subunits[41]. Systemic administration of BFM in WT mice significantly decreased the average daily food intake by ~9% (Fig. 6a), lowered HFD-induced weight gain (Fig. 6b) improved glucose tolerance (Fig. 6c) and insulin sensitivity (Fig. 6g) without changes in glucose-stimulated insulin stimulation (Fig. 6e, f) compared with vehicle controls. To determine whether the effects of BFM on food intake, glucose homeostasis and weight gain were mediated specifically through adipocyte expression of ATP6v0a1, we treated KO mice with BFM. Administration of BFM to KO mice did not significantly effect food intake (Fig. 6a), weight gain (Fig. 6b), glucose tolerance (Fig. 6d), insulin sensitivity (Fig. 6h) and glucose-stimulated insulin secretion (Fig. 6f) compared with vehicle treated KO controls. Therefore, BFM-mediated reduction in food intake, weight gain and improvement in glucose homeostasis is dependent on adipocyte expression of ATP6v0a1. We then measured hypothalamic and WAT gene expression changes in WT mice after 14 days of treatment with BFM or Veh. In a similar manner previously observed in KO mice, BFM-mediated V-ATPase inhibition significantly increased hypothalamic expression of satiety promoting peptides Pomc (by ~3 fold) (Fig. 6i) and WAT expression of Gdf15 (Fig. 6j) compared with Veh-treated mice. In addition, BFM treatment was associated with lower

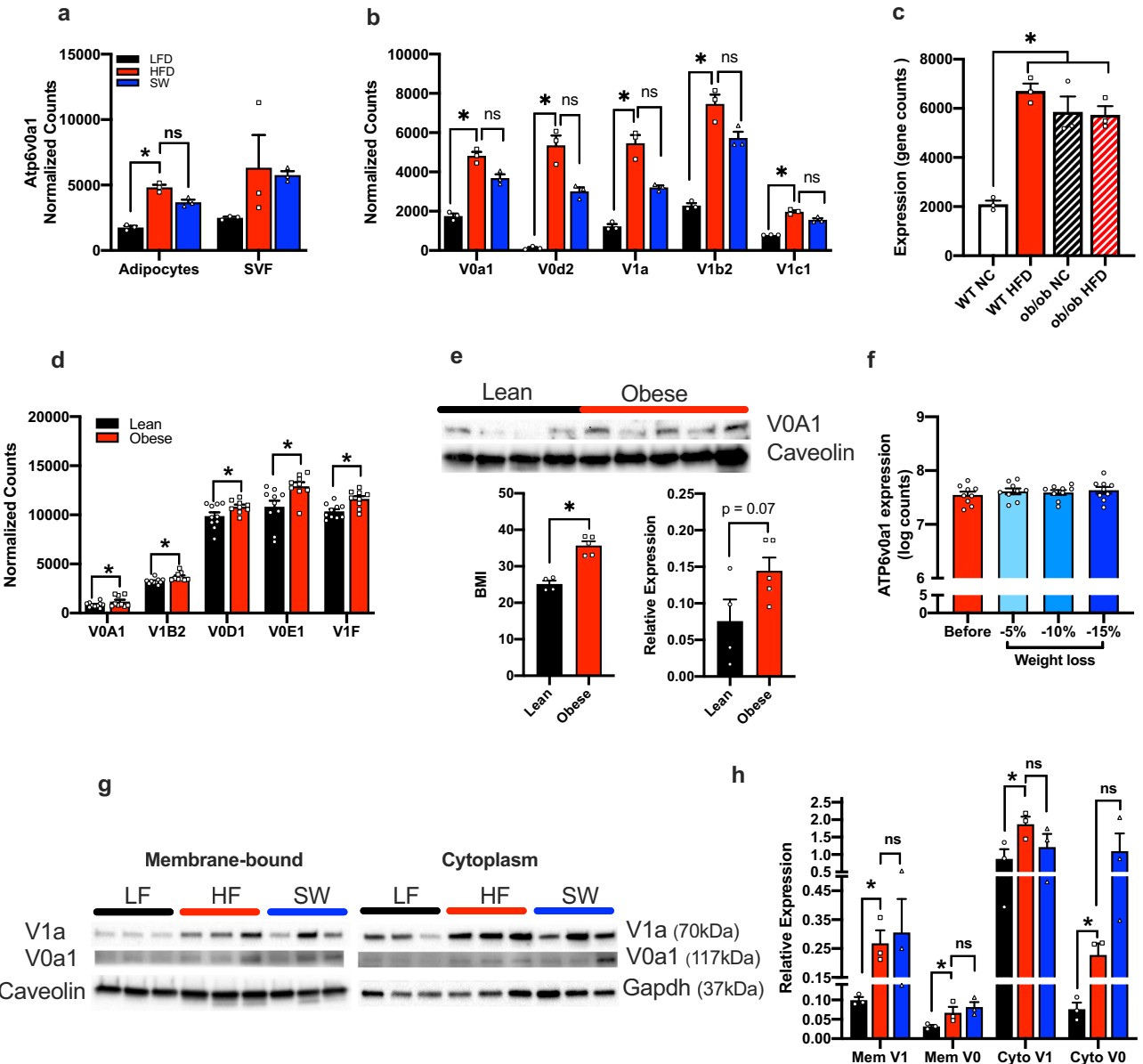

**Fig. 2 | V-ATPase expression in obesity. a** Expression of *Atp6v0a1* in lean (LF, black bars), obese (HF, red bars) and formerly obese (SW, blue bars) mice determined by RNA seq, $n = 3$/group, Adipocyte $p = 0.005$. **b** Expression of V-ATPase subunits in lean, obese and formerly obese mice determined by RNA seq, *denotes differential expression compared to LF fed mice determined as lfdr < 0.05, $n = 3$/group, ($p < 0.004$ for all LF vs HF comparisons). **c** Expression of *Atp6v0a1* in WT (open bars) and ob/ob mice (hatched bars) fed with normal chow (black/white) and high-fat diet (HFD, red) determined by RNA seq (GEO GSE167264,), $n = 3$/group, $p = 0.0002$. **d** Adipocyte gene expression of V-ATPase subunits in lean (black bars) and humans with obesity (red bars), ($n = 9–10$)[27], from Gene Expression Omnibus (GEO: GSE2508)[27] and differential expression between groups determined using GEO2R (default parameters including Benjamini and Hochberg calculation of the false discovery rate, fdr)* = fdr < 0.4 (V0A1 fdr = 0.36, V1B2 = 0.18, V0D1 = 0.35, v0E1 = 0.16, V1F = 0.19). **e** AT6V0a1 expression in subcutaneous adipose tissue from

lean participants ($n = 4$) and individuals with obesity ($n = 5$) (BMI: $t = 6.915$, df = 7, CI = 6.942–14.16, $p = 0.0002$; Atp6V0a1 protein expression: $t = 2.074$, df = 7, CI = −0.009–0.15, $p = 0.07$) was analyzed using 2-tailed student's $t$-test. **f** Adipose gene expression of *ATP6v0a1* in participants with obesity before (red) and after 5%, 10%, and 15% weight loss (shades of blue). Microarray expression data was downloaded from GEO GSE70529[11], ($n = 9$/group). **g, h** Western blot and densinometric quantification of membrane-bound and cytoplasmic Atp6V1a (V1a) and Atp6V0a1 (V0a1) subunits in gonadal WAT from lean (LF), obese (HF), and formerly obese (SW) mice ($n = 3$/group; MemV1: $F = 2.319$, df = 8, $p = 0.09$; MemV0: $F = 4.786$, df = 8, $p = 0.03$; CytoV1: $F = 4.860$, df = 8, $p = 0.03$; CytoV0: $F = 6.324$, df = 8, $p = 0.02$). Densinometric quantification is expressed as mean ± SEM and was analyzed by one-way ANOVA followed by Two-stage linear step-up procedure of Benjamini, Krieger and Yekutieli. * denotes statistical significance at $p < 0.05$.

hypothalamic expression of the pro-feeding peptide (Agrp) compared with Veh treatment (Fig. 6i).

### Inhibition of V-ATPases in formerly obese mice decreased HFD-induced weight regain

To determine whether the persistent increase in the adipocyte expression of Atp6v0a1 plays a role in HFD-induced weight regain, we treated formerly obese mice with BFM (Fig. 7a) while also re-initiating

HFD feeding. As anticipated, we found that V-ATPase inhibition significantly lowered food intake (Fig. 7b), blunted weight re-gain (Fig. 7c) and reduced fat mass (Fig. 7d) compared with vehicle treated mice. The reduced food intake was associated with increased (~2 fold) hypothalamic expression of satiety promoting hormone Pomc (Fig. 7e). Consistent with the previous studies, circulating levels of growth differentiation factor 15 (Gdf15) were significantly elevated (Fig. 7f) and WAT expression of Gdf15 was also increased in BFM

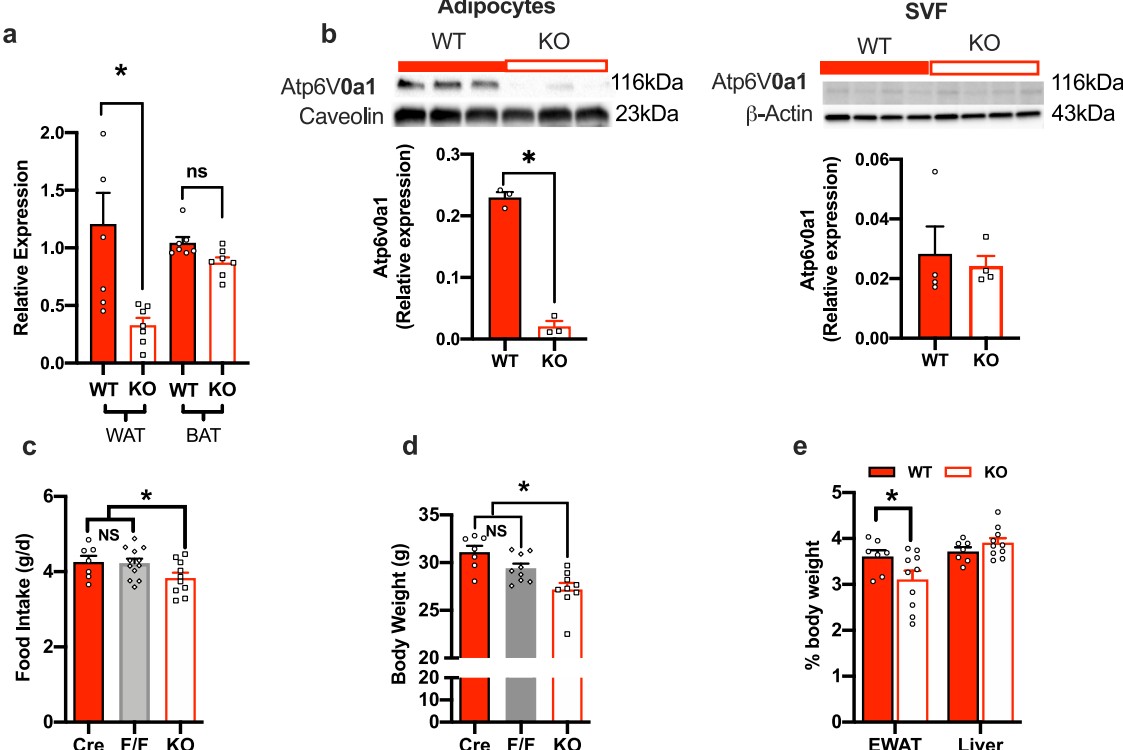

**Fig. 3 | Adipocyte-specific Atp6v0a1 depletion in vitro and in vivo.** *Atp6v0a1* mRNA quantification in WAT and BAT ($f = 7.221$, df = 3, $p = 0.0013$) adipose tissue in KO (white bars) and WT controls (red bars), $n = 7$ per group. **b** Western blots and densinometric quantification of Atp6V0a1 in adipocytes ($t = 16.86$, df = 4, CI = $-0.244$ to $-0.175$, $p < 0.001$) and SVCs ($t = 0.41$, df = 6, CI = $-0.028$ to 0.02, $p = 0.34$) from gonadal WAT from WT (adiponectin Cre) and KO mice ($n = 3-4$/group). **c** Food intake ($F = 3.03$, df = 28, $p = 0.03$), cre, $n = 7$, FF $n = 12$ KO, $n = 10$. **d** Body weight ($F = 9.86$, df = 24, $p < 0.001$), of adiponectin Cre (Cre, $n = 7$) (red bars),

Atp6v0a1$^{fl/fl}$ (F/F, $n = 9$) (gray bars) and KO ($n = 9$, white bars) mice, **e** weight of WAT ($t = 2.416$, df = 30, $p = 0.02$) and liver ($t = 0.88$ df = 30, $p = 0.37$) as % of total body weight of WT and KO mice ($n = 7-12$/group). All mice were fed normal chow. **a, c-e** were analyzed by one-way ANOVA followed by two-stage linear step-up procedure of Benjamini, Krieger and Yekutieli with false discovery rate of 0.10. **b** was analyzed by students *t*-test (2 tailed). Data is expressed as mean ± SEM, *denotes statistical significance at $p < 0.05$.

treated 'rebound' mice compared with vehicle controls (Fig. 7f). BFM also downregulated expression of leptin and lipogenic genes (Fasn, Acaca) and increased Cpt1a, reflective of the decreased fat mass and suggestive of lipid catabolism (Fig. 7g). Furthermore, similar to KO mice, BFM also upregulated WAT expression of classical markers of ER stress including Ddit3 and Atf4 (Fig. 7g). BFM-treated weight rebound mice also had decreased plasma concentrations of C-peptide, insulin, amylin, leptin and resistin compared to VEH-treated weight rebound mice (Supplementary Information, Supplemental Fig. 3c, d). We then measured the subcellular localization of V-ATPase subunits of WAT from BFM and VEH-treated mice. BFM resulted in depletion of both V0a1 ($p = 0.09$) and V1a ($P = 0.00004$) membrane localization (Fig. 7h) compared with Veh-treated mice.

### Atp6v0a1 knockdown in 3T3-derived adipocytes increased levels of Gdf15

To determine if expression of ATP6v0a1 can directly modulate levels of Gdf15 in adipocytes, we conducted a series of experiments in 3T3L1-derived adipocytes. siRNAs targeting Atp6v0a1 resulted in 54% decreased gene expression (Fig. 8a) and 33% decrease in protein levels (Fig. 8b) that resulted in 2-fold increase in Gdf15 at the transcript level (Fig. 8c) and a significant 20% increase in Gdf15 at the protein level (Fig. 8d). No significant changes in leptin or adiponectin were observed. Similar to KO and BFM-treated mice, siRNA treatment increased Atf4 and Ddit3 expression (Fig. 8e). Therefore, lower adipocyte ATP6v0a1 expression results in increased ER stress and increased Gdf-15 levels providing a potential mechanisms underlying adipose-to-brain communication in food intake and body weight regulation (Fig. 8f).

## Discussion

Using a diet switch model in mice, we show that, after weight loss, much of obesity-induced transcriptomic signature persists in the white adipose tissue. The majority of persistently dysregulated genes were localized in adipocytes, but the immune cell fraction (SVCs) also contained a significant fraction of genes that may play also an important role in metabolic memory. We hypothesized that these 'metabolic memory' genes may play a role in establishing a new metabolic set point and driving food intake and weight re-gain. To investigate this, we first conducted a food intake screen in *C. elegans* and found that Atp6v0a1 expression is important in the regulation of food intake. Atp6V0a1 encodes a component of the vacuolar H + ATPase, a large multi-subunit enzyme proton pump that is involved in the regulation of intracellular pH homeostasis[25], but has not previously been linked to food intake or body weight regulation.

Notably, in individuals with obesity, the adipocyte expression of ATP6v0a1 was higher compared with lean participants and expression levels do not decrease after weight loss. To investigate the role of ATP6v0a1 in energy balance, we generated an adipocyte specific ATP6v0a1 KO mice. KO mice gained significantly less weight due to decreased food intake both in normal chow and HFD conditions. To determine if pharmacological inhibition of V-ATPase assembly could modulate energy balance, we administered BFM to mice while initiating HFD. BFM treatment blunted weight gain and food intake and this effect was specifically dependent on adipocyte expression of Atp6v0a1. We then treated formerly obese mice with BFM while initiating HFD re-feeding to induce weight regain and found that inhibition of V-ATPase also decreased food intake and blunted weight regain. Overall, Atp6v0a1 inhibition (via KO or BFM treatment) resulted in

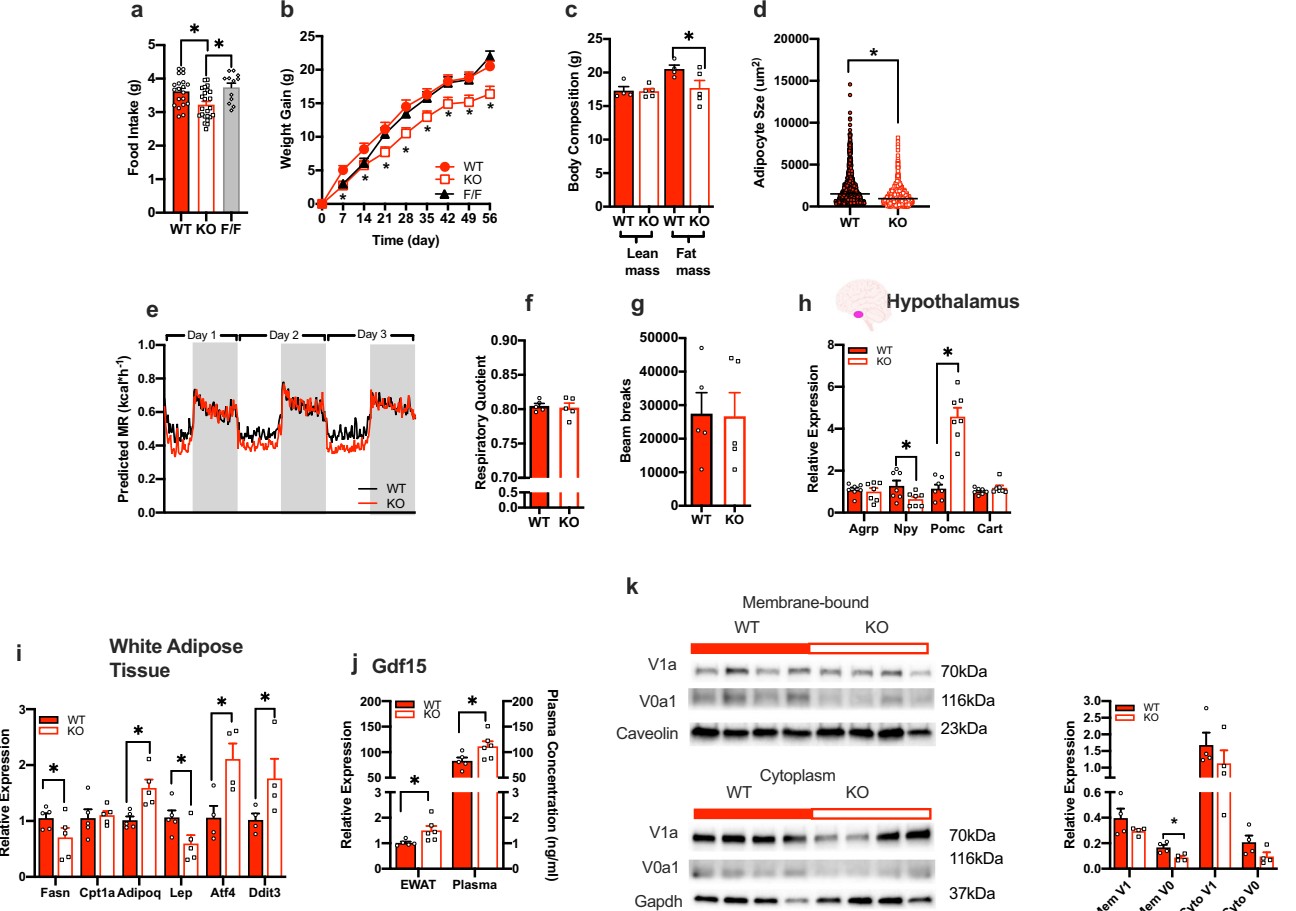

**Fig. 4 | Adipocyte-specific Atp6v0a1 KO mice fed HFD gain less weight due to reduced food intake. a** Average daily food intake (WT $n = 20$, KO $n = 24$, FF $n = 12$, $F = 7.03$, df = 55, $p = 0.002$), **b** weight gain (WT, $n = 13$, KO $n = 16$, FF $n = 12$, $F = 5.06$, df = 266, $p = 0.01$) in WT (red), KO (white) and FF (gray) mice. **c** Body composition ($n = 5$/group, lean mass: $t = 0.11$, df = 7, CI $= -1.70$–1.536, $p = 0.91$; fat mass: $t = 2.07$, df = 7, CI $= -6.06$ to 0.409, $p = 0.04$), **d** adipocyte size ($t = 0.21$, df = 3665, CI $= -1055$ to $-8791$, $p < 0.0001$), $n = 5$ per group, **e** energy expenditure (genotype: Wald Chi-square = 298, df = 1, $p < 0.001$; BW: Wald Chi-square = 7.119, df = 1, $p = 0.08$; time: Wald Chi-square = 1943, df = 372, $p < 0.001$), ($n = 5$/group). **f** Respiratory quotient ($t = 0.345$, df = 8, CI $= -0.02$–0.015, $p = 0.37$), ($n = 5$/group). **g** Spontaneous activity as determined by beam breaks averaged over 3 days ($t = 0.08$, df = 8, CI $= -22429$–20849, $p = 0.47$), in WT and KO fed 60% fat diet for 8 weeks. Quantitative PCR measurement of gene expression within **h** hypothalamus ($n = 7$/group; Agrp: $t = 0.44$, df = 12, $p = 0.33$; Npy: $t = 2.28$, df = 12, $p = 0.02$; Pomc: $t = 7.29$, df = 12,

$p < 0.001$; Cart: $t = 1.30$, df = 12, $p = 0.11$) and **i** gonadal WAT ($n = 5$/group; Fasn: $t = 1.81$, df = 8, $p = 0.05$; Cpt1a: $t = 0.31$, df = 8, $p = 0.38$; Adipoq: $t = 3.55$, df = 8, $p = 0.004$; Lep: $t = 2.34$, df = 8, $p = 0.03$; Atf4: $t = 2.99$, df = 6, $p = 0.01$; Ddit3: $t = 2.01$, df = 6, $p = 0.05$). **j** Gdf15 transcript levels in WAT and plasma protein levels (WT $n = 5$, KO $n = 6$ group; qPCR: $t = 2.51$, df = 9, $p = 0.02$; plasma: $t = 2.27$, df = 9, $p = 0.03$). **k** Western blot and densinometric quantification of membrane-bound and cytoplasmic Atp6v1a (V1a) and Atp6v0a1 (V0a1) subunits in gonadal WAT from WT and KO mice fed with 60% fat diet for 8 weeks ($n = 4$; MemV1: $t = 1.35$, df = 6, $p = 0.23$; MemV0: $t = 3.21$, df = 6, $p = 0.02$; CytoV1: $t = 1.03$, df = 6, $p = 0.34$; CytoV0: $t = 1.80$, df = 8, $p = 0.12$). **a, b** were analyzed by one-way ANOVA followed by two-stage linear step-up procedure of Benjamini, Krieger and Yekutieli with false discovery rate of 0.10. **c, d, f–k** were analyzed by students $t$-test (2 tailed). **e** was analyzed using ANCOVA with body weight as covariate using SPSS v22. Data is expressed as mean ± SEM, * denotes statistical significance at $p < 0.05$.

consistently lower food intake, ranging from 10–15% compared to their respective control groups resulting in lower body weight, and lower HFD-induced weight gain.

V-ATPases are ATP-dependent proton pumps that function to acidify a wide array of intracellular compartments by pumping protons across the plasma membranes of various cell types[25] to activate enzymes, dissociate hormones from receptors and degrade internalized macromolecules[42,43]. ATPases have crucial functions in many pathophysiological processes making them prime targets in the development of therapies for diseases such as osteoporosis and cancer[44]. This proton pump is composed of a cytosolic $V_1$ domain (subunits A-H) and a transmembrane $V_0$ domain (subunits a, c, c', c'', d, e) and is highly conserved in many species including yeast, *C. elegans*, mouse and human[25,26,31]. An important mechanism by which V-ATPase activity is controlled is the reversible assembly of its two domains whereby Atp6V1 translocates from the cytoplasm and associates with membrane-bound Atp6V0 subunit[45]. Cells can adjust V-ATPase assembly, activity,

and membrane distribution in response to various acute stimuli including relative glucose levels within the cell[32,45–47]. In obesity, we observed increased expression of both domains and increased co-localization of the V1 domain with V0, which both contribute to overall increased V-ATPase activity and this increased activity persisted after weight loss. Furthermore, membranal expression of the V0 subunits were specifically reduced in KO mice and by BFM treatment compared with controls. Therefore, suggesting that the persistent obesity-induced increase in V-ATPase activity plays a key role in weight re-gain.

V-ATPases are found in many intracellular compartments including endosomes and Golgi and have a well-described role in lysosomes[48,49]. Lysosomal V-ATPase interacts with the Rag-regulator complex and mTOR the serine/threonine kinase that plays a pivotal role in nutrient sensing[50] and maintaining an equilibrium between anabolic and catabolic processes[51]. In starvation conditions, autophagy provides an internal source of nutrients for energy generation[52] and survival[53] via degradation of intracellular components such as proteins,

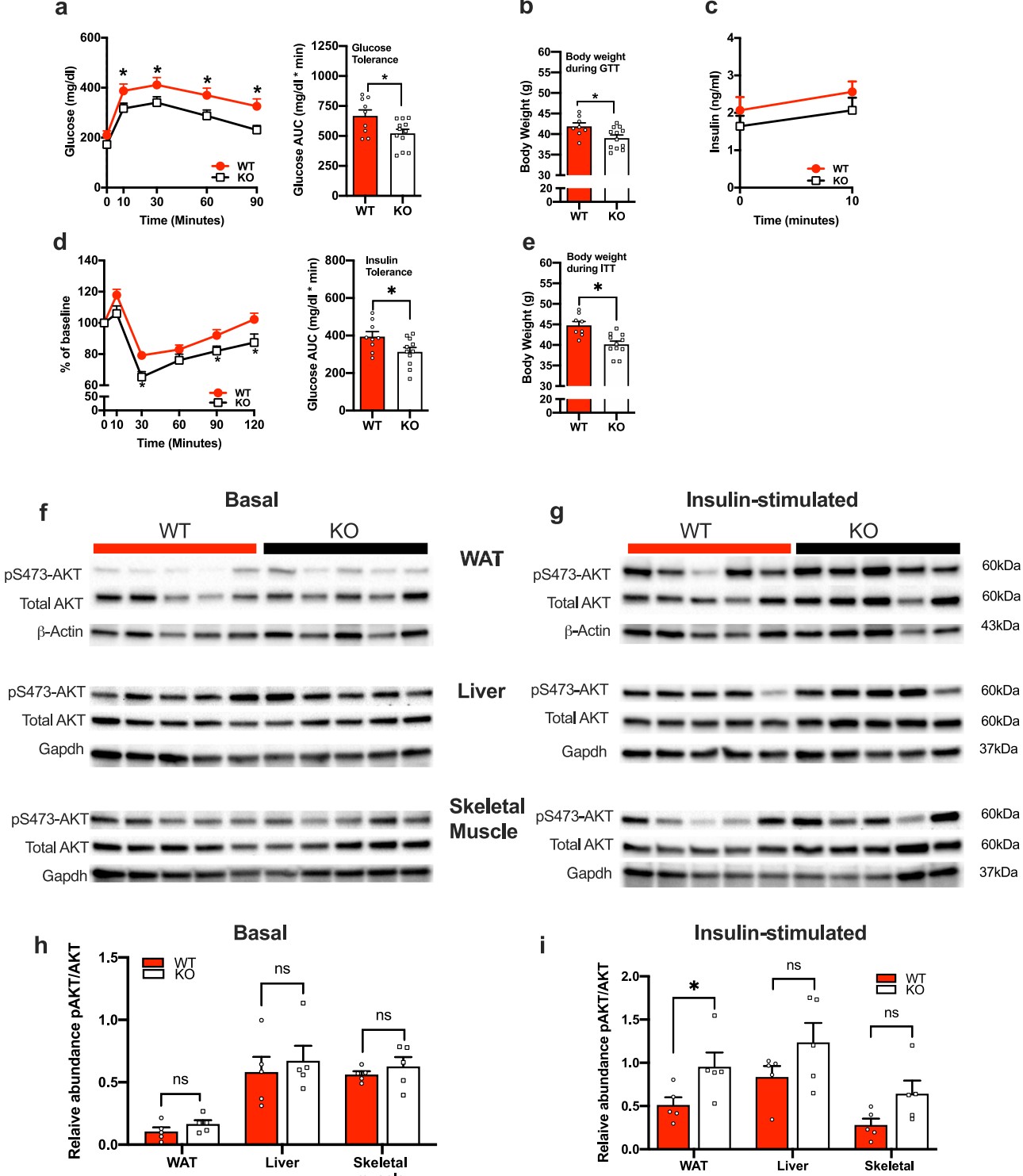

**Fig. 5 | ATP6v0a1 depletion in adipocytes results in improved glucose homeostasis. a** Glucose tolerance test (GTT; genotype: $F = 6.15$, df = 1, $p = 0.02$; time: $F = 63.69$, df = 4, $p < 0.001$; genotype X time: $F = 1.40$, df = 4, $p = 0.24$). **b** Body weight at time of GTT ($t = 2.52$, df = 19, $p = 0.01$). **c** Glucose-stimulated plasma insulin (genotype: $F = 1.14$, df = 1, $p = 0.30$; time: $F = 11.37$, df = 1, $p = 0.003$; genotype X time: $F = 0.06$, df = 1, $p = 0.80$). **d** Insulin tolerance test (ITT), presented as change from baseline (genotype: $F = 6.76$, df = 1, $p = 0.01$; time: $F = 59.95$, df = 5, $p < 0.001$; genotype X time: $F = 2.18$, df = 5, $p = 0.06$). **e** Body weight at time of ITT, ($n = 9$-$12$/group; $t = 3.78$, df = 17, $p < 0.001$). Immunoblots of total AKT and phosphorylated AKT at Ser473 in WAT, liver and skeletal muscle in **f** the basal state **g** insulin stimulated state, of WT and KO mice fed with 60% fat diet for 8 weeks ($n = 4$/group). **h** Densinometric quantification of basal blots (WAT: $t = 1.35$, df = 8, $p = 0.11$; Liver: $t = 0.53$, df = 8, $p = 0.30$; Skeletal muscle: $t = 0.81$, df = 8, $p = 0.22$). **i** Densinometric quantification of insulin stimulated blots (WAT: $t = 2.37$, df = 8, $p = 0.04$; Liver: $t = 1.55$, df = 8, $p = 0.16$; Skeletal muscle: $t = 2.15$, df = 8, $p = 0.06$). WT (red circles, red bars), KO (white squares, open bars). Data is expressed as mean ± SEM. **a**, **c**, **d** were analyzed by repeated measures two-way ANOVA followed by two-stage linear step-up procedure of Benjamini, Krieger and Yekutieli with false discovery rate of 0.10. **b**, **e**, **h**, **i** were analyzed by student $t$-test (2 tailed). * denotes statistical significance at $p < 0.05$.

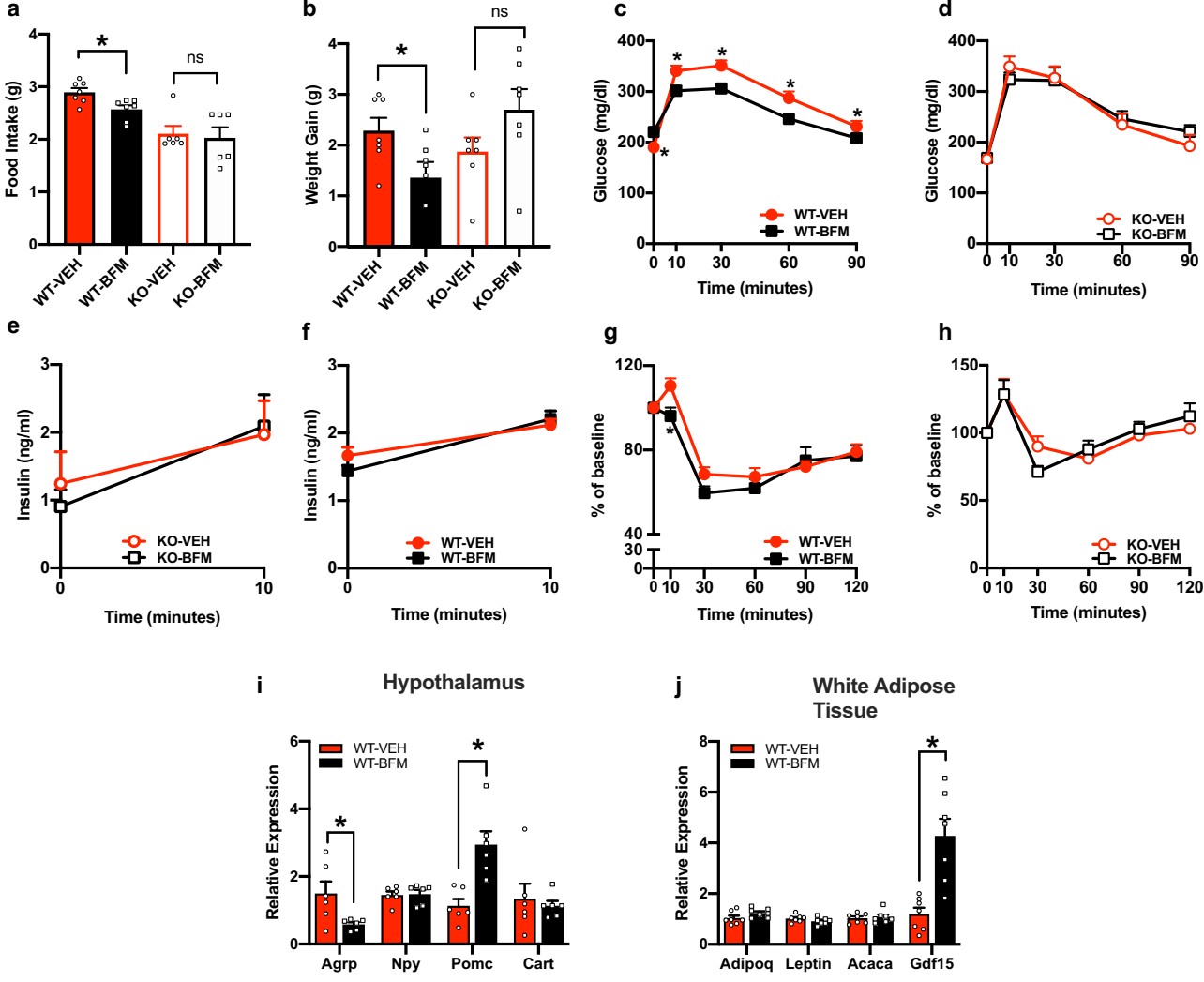

**Fig. 6 | Inhibition of V-ATPase abrogates HFD-induced weight gain which is dependent on the adipocyte expression of Atp6v0a1. a** Average daily food intake ($F = 10.21$, df = 3, $p < 0.001$), (WT-VEH $n = 7$, WT-BFM $n = 7$, KO- VEH $n = 6$, KO-BFM $n = 6$). **b** weight gain in WT (solid bars) and KO (open bars) mice after 7 days treated with BFM (black) or VEH (red) and fed HFD ($F = 3.27$, df = 3, $p = 0.03$), (WT-VEH $n = 7$, WT-BFM $n = 7$, KO-VEH $n = 6$, KO-BFM $n = 6$). **c, d** GTT after 3 days of treatment (group: $F = 1.19$, df = 3, $p = 0.33$; time: $F = 206$, df = 4, $p < 0.001$; genotype X time: $F = 4.48$, df = 12, $p < 0.001$), (WT-VEH $n = 8$, WT-BFM $n = 8$, KO-VEH $n = 7$, KO-BFM $n = 8$). **e, f** Glucose stimulated insulin secretion, (group: $F = 0.35$, df = 3, $p = 0.78$; time: $F = 38.62$, df = 1, $p < 0.001$; genotype X time: $F = 1.46$, df = 3, $p = 0.25$), (WT-VEH $n = 8$, WT-BFM $n = 8$, KO-VEH $n = 7$, KO-BFM $n = 8$). **g, h** ITT after 7 days treatment with BFM (WT-VEH $n = 8$, WT-BFM $n = 7$, KO-VEH $n = 8$, KO-BFM $n = 7$: $F = 8.02$,

df = 3, $p = 0.006$; time: $F = 74.23$, df = 5, $p < 0.001$; genotype X time: $F = 4.25$, df = 15, $p < 0.001$). Relative gene expression in **i.** hypothalamus ($n = 6$/group; Agrp: $t = 2.506$, df = 10, $p = 0.03$; Npy: $t = 0.17$, df = 10, $p = 0.86$; Pomc: $t = 4.055$, df = 10, $p = 0.002$; Cart: $t = 0.45$, df = 10, $p = 0.66$) and **j** WAT ($n = 7$/group; Adipoq: $t = 1.71$, df = 12, $p = 0.11$; Lep: $t = 1.39$, df = 12, $p = 0.19$; Acaca: $t = 0.41$, df = 12, $p = 0.69$; Gdf15: $t = 4.33$, df = 12, $p = 0.001$) in vehicle or BFM-treated WT mice after 2 weeks of treatment. Data is expressed as mean ± SEM. **a**, **b** were analyzed by one-way ANOVA followed by two-stage linear step-up procedure of Benjamini, Krieger and Yekutieli with false discovery rate of 0.10. **c**–**h** were analyzed by repeated-measures two-way ANOVA followed by two-stage linear step-up procedure of Benjamini, Krieger and Yekutieli with false discovery rate of 0.10. **i, j** were analyzed by student $t$-test (2 tailed), * denotes statistical significance at $p < 0.05$.

and glycogen which yields amino acids and glucose, respectively. In effect, the lysosome acts as a metabolic sensor and orchestrates the transcriptional response to fasting[54,55]. The cell can sense the luminal levels of lysosomal amino acids and convey this information to the cytosolic side via the V-ATPase[50]. Upon starvation, the decreased concentration of amino acids inside the lysosome is sensed by the V-ATPase and results in decreased mTORC1 activity and the induction of autophagy[50]. Importantly, in obesity, autophagy is reduced leading to intracellular storage of unwanted material and lipid accumulation[56,57] and impaired lysosomal function[58]. Moreover, V-ATPases play an important role in nutrient sensing and obesity-induced changes in their expression and function persist after weight loss making the cell less sensitive to nutrient fluctuations. Furthermore, perturbation in the ability of the V-ATPases to relay nutritional information is likely to be an important contributor to weight regain.

Our studies utilized a HFD (60% calories from fat) to induce obesity. The 60% HFD diet is commonly used in mouse obesity research as it accelerates weight gain and results in significant increase in fat mass. However, recent discussion on the topic of high-fat diet composition in relation to the human diet highlight the importance of consideration of macronutrient content in the context of mimicking human dietary habits[59]. Our data shows ATP6v0a1 expression is also elevated in genetic models of obesity and adipocytes from human individuals with obesity suggesting the effects are independent of diet composition.

It is well established that there is crosstalk between the adipose tissue and hypothalamus[60]. The communication between these tissues is mediated by integration of the sympathetic nervous system and circulating factors[60,61]. Elevated circulating Gdf15 has a well-described role in the regulation of food intake, with elevated levels

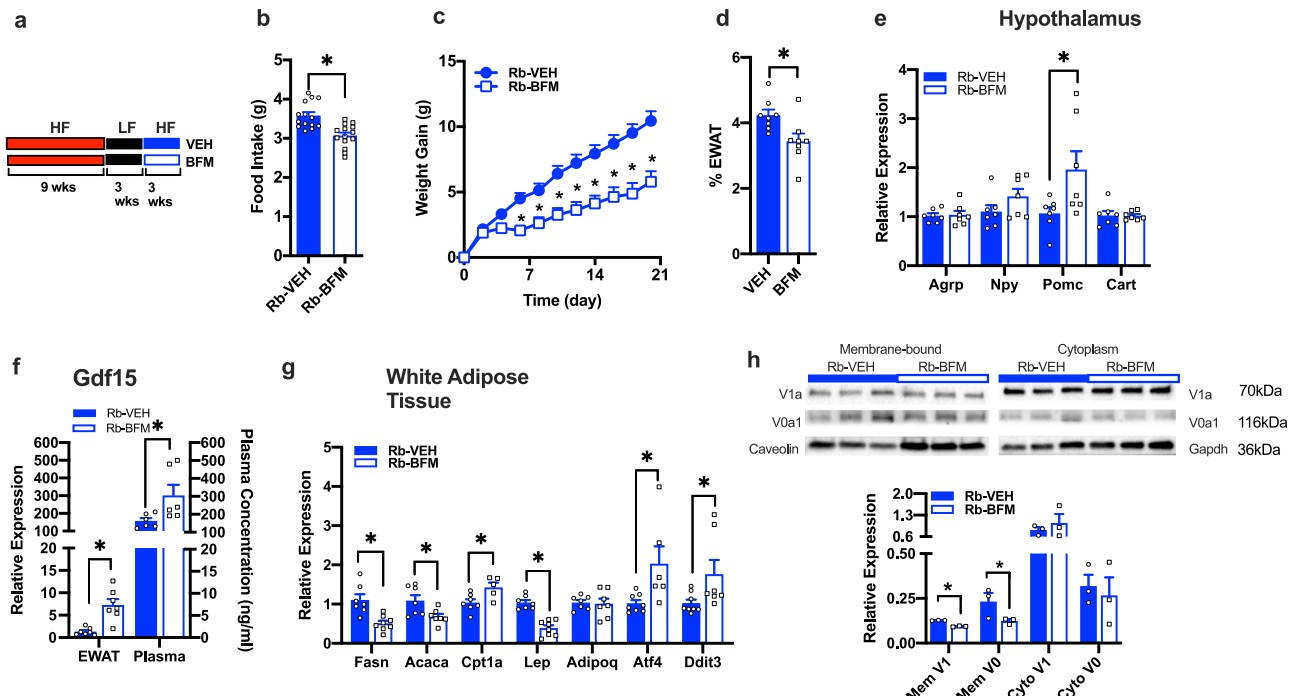

**Fig. 7 | Inhibition of V-ATPase abrogates HFD-induced weight re-gain in formerly obese mice. a** Schematic diagram of weight rebound mouse experiment. **b** Average daily food intake ($t = 4.31$, df = 26, $p < 0.001$) and **c** HFD-induced weight re-gain ($n = 14$/group; group: $F = 18.29$, df = 1, $p = 0.002$; time: $F = 119$, df = 10, $p < 0.001$; genotype X time: $F = 13.78$, df = 10, $p < 0.001$). **d** WAT as percentage of body weight ($n = 8$; $t = 2.74$, df = 14, $p = 0.01$). **e** Relative gene expression in hypothalamus as determined by qPCR (Agrp: $t = 0.18$, df = 12, $p = 0.43$; Npy: $t = 1.53$, df = 12, $p = 0.15$; Pomc: $t = 2.27$, df = 12, $p = 0.04$; Cart: $t = 0.04$, df = 12, $p = 0.96$). **f** Gdf15 gene levels in WAT ($n = 7$/group; $t = 4.46$, df = 12, $p < 0.001$) and plasma concentration ($n = 6$/group; $t = 2.32$, df = 10, $p = 0.04$). **g** Relative gene expression in WAT (Fasn: $t = 3.93$, df = 13, $p = 0.001$; Acaca: $t = 2.58$, df = 12, $p = 0.02$; Cpt1a:

$t = 2.73$, df = 10, $p = 0.02$; Lep: $t = 6.74$, df = 13, $p < 0.001$; Adipoq: $t = 0.21$, df = 12, $p = 0.84$; Atf4: $t = 2.86$, df = 12, $p = 0.02$; Ddit3: $t = 2.11$, df = 13, $p = 0.05$). **h** Western blot and densinometric quantification of membrane-bound and cytoplasmic Atp6V1a and Atp6V0a1 in gonadal WAT ($n = 3$/group; MemV1: $t = 19.18$, df = 4, $p < 0.001$; MemV0: $t = 2.22$, df = 4, $p = 0.04$; CytoV1: $t = 0.76$, df = 4, $p = 0.25$; CytoV0: $t = 0.43$, df = 4, $p = 0.34$) of WT mice treated with either Veh (Blue) or BFM (white) during weight rebound (Rb). Data is expressed as mean ± SEM and * denotes statistical significance at $p < 0.05$. **b**, **d**–**h** were analyzed by student $t$-test (2 tailed), **c** was analyzed by repeated-measures two-way ANOVA followed by two-stage linear step-up procedure of Benjamini, Krieger and Yekutieli with false discovery rate of 0.10.

of Gdf15 driving decreased food intake in both mouse[62,63] and human studies[64,65]. In our studies, Atp6v0a1 inhibition in vitro and in vivo was associated with increased adipose tissue expression and elevated circulating Gdf15 levels. Recent literature has highlighted a mechanistic link between ER stress and Gdf-15 levels[39,40]. Notably, C/EBP homologous protein (Chop, Ddit3) has been shown to directly bind to the promoter of GDF15 and activates its transcription under ER stress conditions[39,40]. Our studies revealed that Atp6V0a1 inhibition increased the expression of Chop, potentially linking Atp6v0a1 with Gdf-15. Furthermore, reduced food intake in both HFD-fed KO and HFD fed-BFM treated weight rebound mice was associated increased hypothalamic expression of Pomc compared with respective controls. Gdf15 has been shown to regulate neuropeptide levels in the hypothalamus whereby systemic or direct injection of Gdf15 to the in the arcuate nucleus increased Pomc expression[62]. Therefore, we propose a potential link between adipose V-ATPase and key appetite regulators including Gdf15 and Pomc (Fig. 8f).

In summary, using a multi-species approach, we identified obesity-induced adipocyte genes that persist after weight loss and likely contributes to weight re-gain by enhancing food intake. We described a role for adipocyte ATP6v0a1 in the regulation of food intake and body weight. Expression of V-ATPase subunits in adipocytes increases in the obese state and persists despite weight loss in both mice and humans. Furthermore, the molecular and pharmacological disruption of V-ATPase assembly in adipocytes potentially modulates the adipose-to-brain axis communication leading to both decreased food intake, weight loss and blunted weight re-gain. Thus, ATP6v0a1 is

a target for future anti-obesity therapeutic strategies and the peripheral origin of this approach may reduce the likelihood of interfering with mental health than anti-obesity strategies that act directly on the brain.

## Methods

All experiments were approved by and conducted in accordance with the University of California, San Diego IACUC. All animals were housed with a 12:12 light–dark cycle, temperature 20–21 °C, and average 50% humidity.

### Diet switch model

Male wild type C57BL/6 J mice (Jackson Laboratories) were group housed in standard cages and fed either a 10% (LFD, D12450, Research Diets, New Brunswick, NJ) or 60% fat diet (HFD, D12492) from 9–10 weeks of age. After 9 weeks of HFD feeding, half of the HFD-fed mice where switched to the LFD (SWD) for 3 weeks to induce ~10% weight loss which is sufficient to normalize glucose tolerance and insulin sensitivity to equivalent levels in observed in lean, normal chow fed mice[10,66] (Fig. 1a). At the end of the study mice were sacrificed, epididymal white adipose (WAT) dissected and fractionated into adipocytes and SVCs or stored at −80 °C until further analyses.

### cDNA library generation, RNA-sequencing and analysis

Adipocytes and stromal vascular cells (SVCs) were prepared from adipose tissue[67]. Briefly, epididymal fat pads were weighed, rinsed three times in PBS, and then minced in FACS buffer (PBS + 1% low

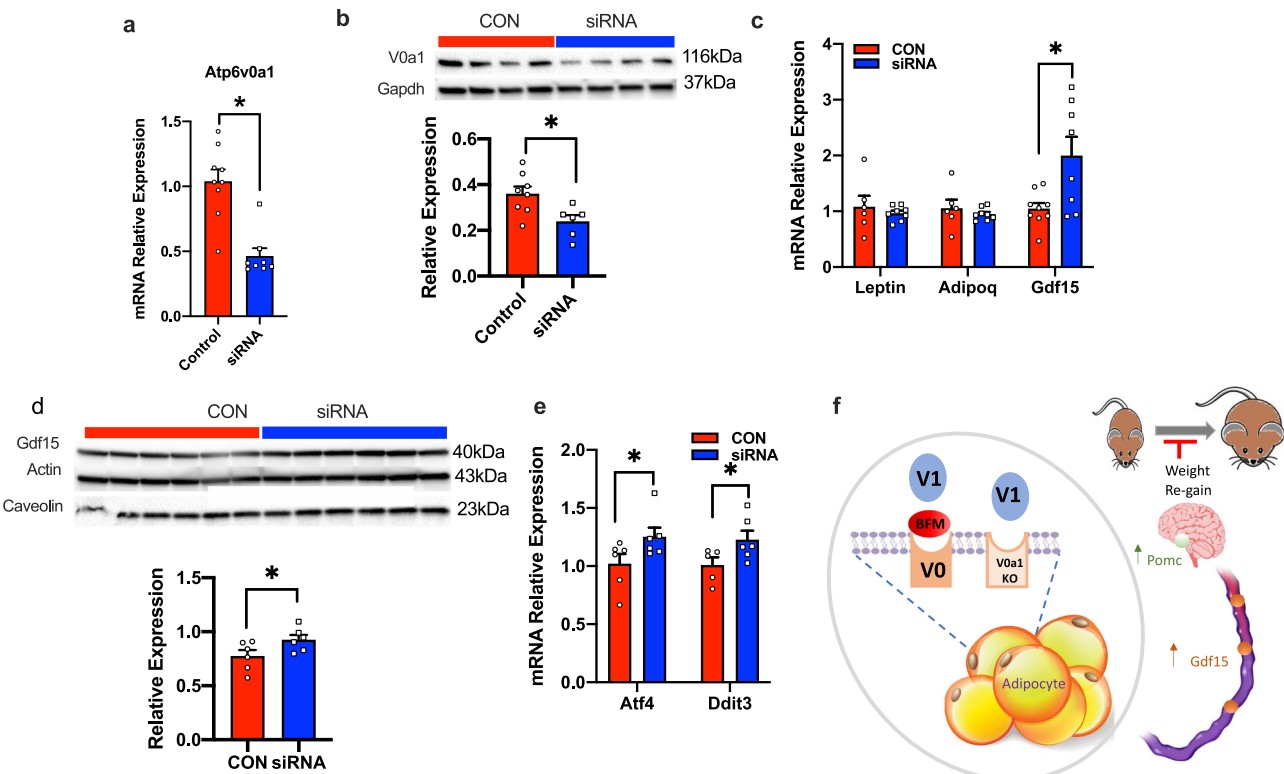

**Fig. 8 | Adipocyte-specific knockdown of Atp6v0a1 increases Gdf-15 expression in 3T3L1 adipocytes. a** Relative *Atp6v0a1* gene expression in 3T3-derived adipocytes treated with SiRNA targeting *Atp6v0a1* (blue) or control siRNA (red) (control $n = 9$, SiRNA $n = 8$/group; $t = 5.10$, df = 15, $p < 0.001$). **b** Western blots and densinometric quantification of ATP6v0a1 (control, $n = 8$, SiRNA $n = 6$/group; $t = 2.78$, df = 12, $p = 0.008$). **c** Relative gene expression of leptin, adiponectin and Gdf-15 (Lep: $t = 0.67$, df = 12, $p = 0.51$; Adipoq: $t = 0.76$, df = 12, $p = 0.46$; Gdf15: $t = 2.86$, df = 12, $p = 0.01$), control, $n = 6$, SiRNA $n = 9$/group. **d** Western blots and densinometric quantification of Gdf15 ($n = 6$/group; $t = 2.15$, df = 10, $p = 0.03$). **e** Relative gene expression of Atf4 and Ddit3 ($n = 6$/group; Atf4: $t = 2.15$, df = 19, $p = 0.02$; Ddit3: $t = 1.94$, df = 19, $p = 0.04$). **f** Schematic representation on how the inhibition of adipocyte V-ATPase activity by bafilomycin or genetic deletion of Atp6v0a1 leads to increased circulating Gdf15 levels that is associated with increased Pomc expression, reduced food intake and blunted weight gain/re-gain. Data is expressed as mean ± SEM and was analyzed using students *t*-tests (2 tailed), * denotes statistical significance at $p < 0.05$.

endotoxin BSA). Tissue suspensions were treated with collagenase (1 mg/ml, Sigma-Aldrich) for 30 min, and then were filtered through a 100 µm filter (BD Biosciences). After centrifugation at 500 *g* for 5 min, the supernatant containing adipocytes was removed and the pallet containing SVCs fraction was incubated with RBC lysis buffer (eBioscience) for 5 min followed by another centrifugation (300 *g*, 5 min) and resuspension in FACS buffer. Total RNA was isolated using TRIzol (Invitrogen) according to the manufacturer's instructions. The quality of the RNA was assessed using the Tapestation 2200 (Agilent) and all samples passed an RNA Integrity score (RIN) above 7.5 denoting good quality RNA. Libraries were prepared using TruSeq Library prep kits (Illumina) and run on the Hiseq 2500 (Illumina) to obtain approximate coverage of 10 million reads per sample. Reads generated by the sequencer (PE 100) were first mapped to the mouse transcriptome using *Bowtie2* algorithm[68] and counted as reads per gene using RSEM[69] and then analyzed using the statistical algorithm limma[70]. Genes were then sorted by their posterior error probability (also called local false discovery rate, (lfdr)[71]) in a three-way comparisons between conditions LFD, HFD and SWD. In order to detect genes significantly changed by HFD diet that do not revert to their LFD levels after switch (irreversible genes), we look for genes for which the expression (HFD-LFD)−(SWD-HFD) is significantly different from zero. In technical language, we chose contrasts of the test to be [2,−1,−1]. The RNA-seq data generated in this study have been deposited in the GEO database under accession code GSE210014. The processed RNA data generated in this study are provided in the Supplementary Data 1, 2.

### Food intake assay in *C. elegans*

Worm mutants were obtained from the Caenorhabditis Genetics Center (CGC). Food intake in *C. elegans* was measured in age-synchronized worms that were seeded into 96 well plates as L1 larvae in the presence of OP50 bacteria (6 mg/ml final conc.) as a food source and maintained at 20 °C. At the L4 stage, they were sterilized with 5-fluoro-2′-deoxyuridine (FUDR) (0.12 mM final). Food intake was measured by determining the amount of bacteria in each well using the absorbance at 600 nm ($OD_{600}$) and its depletion over the first 4 days of adulthood. The number of worms in each well was than counted under an inverted microscope to normalize bacterial consumption per worm. To account for experimental variation between experiments, all food intake values were normalized to food intake of wild type N2 worms which was set 100%[20].

### Gene expression data

Adipose expression of ATP6v0a1 in WT and leptin deficient (ob/ob) mice was obtained from expression array data (GEO GSE167264). Human adipocyte microarray expression data from lean individuals and participants with obesity ($n = 9$–10) were downloaded from Gene expression omnibus (GEO: GSE2508)[27]. GEO2R (using default parameters including Benjamini and Hochberg calculation of the False discovery rate) was used to compare differential expression between lean individuals and participants with obesity. Human adipose microarray data was obtained from individuals with obesity before and after 5%, 10%, and 15% weight loss ($n = 9$ per group), GEO GSE70529[11], and expression of ATP6v0a1 was determined in each group. Primer sequences are listed in Supplemental Data 7.

## Human adipocyte studies

Subcutaneous adipose tissue samples were obtained from the lateral abdominal wall region (peri-umbilical) after an overnight (10–12 h) fast[72] from non-diabetic participants. Adipocytes isolation and protein extraction were performed as described previously[73]. Participants were classified as non-diabetic on the basis of values of either HbA1c < 5.7 or 2 h [glucose] < 140 mg/dL within 2 months of biopsy. They also had to be negative for a family history (1st degree relative) of type 2 diabetes. The following lean individuals (BMI/sex/age, 26.9/M/37, 23.7/M/40, 23.5/F/60, 26.5/F/46) and individuals with obesity (BMI/sex/age, 38.2/F/52, 33.1/M/37, 38/M/53, 32.9/F/47, 36.3/M/64) were studied. The Committees on Human Investigation of the University of California, San Diego, and VA San Diego Healthcare System approved the experimental protocol. Informed written consent was obtained from all participants after explanation of the protocol.

## Generation of mice with adipocyte-specific ablation of Atp6v0a1 (knockout, KO)

Atp6v0a1$^{fl/fl}$ mice (on a C57BL/6 J background) were generated by conducting IVF using sperm from the Toronto Centre for Phenogenomics (TCP). These mice were then crossbred with adiponectin cre (stock number 028020, Jackson Labs) mice to generate adipocyte-specific ablation of Atp6v0a1 (KO). Adiponectin-cre mice were used as controls and in figures are referred to as 'WT' for simplicity. KO mice and WT controls were studied from 10 weeks of age. Mice were fed normal chow (Lab diet, #5001) and food intake was measured daily for 7 day consecutive days. Body weight was measured at sacrifice where WAT was processed for cellular fractionation as described above.

## HFD studies

Ten-week old male KO and WT control mice were fed 60% HFD. Food intake was measured daily during the first 7 days and then weekly while body weight was measured weekly for 8 weeks. Lean and fat mass was determined using DEXA-scan (Lunar Piximus, Madison, WI) in anesthetized mice. On the day of sacrifice, mice were fasted for 6 h, and blood, WAT and hypothalamus sampled and stored at −80 °C until further analyses.

## Indirect calorimetry

Seven-week HFD-fed mice were acclimatized to metabolic chambers (CLAMS, Columbus Instruments, Columbus, OH) for 3 days before data collection. Food intake, respiratory quotient, and locomotor activity (beam breaks) were continuously monitored and averaged across 3 days. Data was processed using CLAMS data eXamination (CLAX) software (Columbus Instruments).

## Quantitative Real-Time PCR

RNA was isolated from the WAT and hypothalamus, and purified using Trizol and commercial isolation kits (Qiagen RNA easy). WAT and BAT gene expressions were normalized to housekeeping genes Hprt1 and Atp5e[74] while the hypothalamic gene expressions were normalized to housekeeping Hprt1 and Pgk1[75].

## Plasma biochemical and hormone assays

Terminal blood samples were collected via intracardiac puncture in tubes containing 0.5 M EDTA, protease inhibitors, DPP-IV inhibitor and 4-benzenesulfonyl fluoride hydrochloride. Plasma was separated by spinning at 4500 g for 15 min and later stored at −80 °C until analysis. Plasma insulin was measure using 900-MPMI-02 (Alpco, NH, USA), gut-derived and inflammatory peptides using MADKMAG-71K (Millipore, MA, USA) and Gdf15 using MDG150 (R&D Systems, MN, USA).

## Protein fractionation and immunoblotting

Total WAT and liver proteins were extracted using NP-40 with 0.03 M phenylmethylsulfonyl fluoride and protease inhibitor cocktail. Membrane-bound and cytoplasmic WAT proteins were isolated using ProteoExtract Native Protein kit as recommended by the manufacturer. Proteins (5-20 ug) were fractionated in 4–15% Mini PROTEAN TGX acrylamide gels, transferred to PVDF, blocked with 5% BSA, incubated with the primary antibody overnight and secondary antibody for 60 min before detection using ECL (SuperSignal, Thermo Fisher 34580, 34095). Band intensities where quantified using densinometer in ImageLab. Total and pSer473 AKT were normalized to ß-actin or Gapdh, membrane-bound protein to caveolin and cytoplasmic proteins to Gapdh or ß-actin. Sources and dilution of antibodies used are as follows: Total AKT (1:1000, Cell Signaling 4691), pSer473-AKT (1:2000, Cell Signaling 4060), ß-actin (1:1000, Cell Signaling 3700), caveolin (1:1000, BD Transduction Lab 610407), Gapdh (1:2000, Cell Signaling), Atp6v0a1 (1:1000, Antibody Verify AAS81465C), Atp6v1a (1:1000, Abcam 199325), Gdf15 (1:100, Santa Cruz 377195) Anti-mouse IgG (1:4000, Jackson Immoresearch 115035003), Anti-rabbit IgG (1:4000, GE Healthcare NA934V). All uncropped blots are provided in associated 'source data' file.

## Glucose tolerance test (GTT) and Insulin tolerance tests (ITT)

Mice were fasted for 6 h and injected IP with dextrose (1 g/kg, Hospira, Lake Forest, IL) for GTTs or with insulin (0.7 U/kg, Novolin R, Novo-Nordisk, Princeton, NJ) for ITTs. Blood was drawn at 0, 15, 30, 60, 90, and 120 min after injection for blood glucose determination using an Easy Step blood glucose monitor (Home Aid Diagnostics Inc, Deerfield Beach, FL).

## Acute insulin administration

Male 8-week HFD-fed KO and WT mice were anesthetized and basal liver and WAT biopsies were obtained before administration of 0.7 mU/kg insulin through the portal vein. Insulin stimulated tissue biopsies of the liver, muscle and WAT were collected at 3, 5 and 7 min, respectively, and then frozen at −80 °C for future analyses.

## V-ATPase inhibitor treatment

Eleven-week old male WT C57BL/6 J or KO mice were individually housed and acclimatized to IP injections with 0.9% NaCl for 3 days and then randomized to receive daily bafilomycin B1 (100 nmol/kg)[76] or vehicle (5% DMSO in 0.9% NaCl) injections ~1 h before the onset of dark period. GTT was performed on day 3 and an ITT on day 7 after a 6 h fast. Food intake was measured daily and body weight was measured every 2 days. All mice were treated for 14 days and then sacrificed, WAT and hypothalamus dissected and frozen at −80 °C for future analyses.

## Weight re-bound model

Nine-week old male C57BL/6 mice were fed HFD for 9 weeks and were switched to a LFD for 3 weeks. The mice were switched back to HFD and then randomized to receive IP injections for either bafilomycin B1 (100 nmol/kg)[76] or vehicle (5% DMSO in 0.9% NaCl) ~1 h before the onset of dark period. Food was measured daily while body weight was measured every 2 days for 3 weeks. On the day of sacrifice, mice were fasted for 6 h and was administered vehicle or bafilomycin B1 and were euthanized 1 h post-injection. Blood, WAT and hypothalamus were collected, processed and samples frozen at −80 °C for future analyses.

## 3T3-L1 in vitro studies

3T3-L1 cells (ATCC, VA, USA) were grown and maintained in high-glucose DMEM supplemented with 10% fetal bovine serum, L-glutamine (Cat. 25030081, Gibco, NY), and 10 u/ml of penicillin and 10 ug/ml of streptomycin (Cat. 15149-122, Gibco) of in a 5% $CO_2$ environment. The cells were allowed to grow until 70–80% confluency and then differentiated by the addition of isobutylmethylxanthine (500 µM), dexamethasone (25 µM), and insulin (4 µg/ml) for 10 days. The medium was then changed every 2 days. Transient transfection of

3T3-L1 cells were performed in a six-well plate with 30 nM Atp6v0a1 siRNA on target plus SMARTpool (L-040215-01) or negative SMARTpool control (D-001810-10, Horizon Discovery, Cambridge, UK). Cells were transfected with Lipofectamine RNAiMAX Transfection Reagent (Cat. 13778030, Invitrogen, Carlsbad, CA) as described in the manufacturer's protocol. Twenty-four hours after transfection, the growth medium was replaced and cells grown for another 48 h before cells were harvested and processed for both RNA and protein quantification.

## Statistics

All data are expressed as mean ± standard error of mean (SEM). Biological replicates are indicated in the figure legends. Statistics were carried out using GraphPad 8.1. Normal distribution was tested using Shapiro-Wilk test prior to proceeding with an unpaired one-tail Student's $t$-test, one-way ANOVA or a two-way ANOVA with repeated measures followed by post hoc tests involving two-stage linear step-up procedure of Benjamini, Krieger and Yekutieli corrections for multiple comparisons with a false discovery rate of 0.10, whenever applicable. Energy expenditure was analyzed by ANCOVA using body weight as covariate in SPSS v22[77]. Statistical significance was set at $\alpha = 0.05$.

## Reporting summary

Further information on research design is available in the Nature Research Reporting Summary linked to this article.

## Data availability

Additional gene expression data are provided in Supplemental Data 1, 2. The RNA-seq data generated in this study have been deposited in the GEO database under accession code GSE210014. The processed RNA data generated in this study are provided in the Supplementary Data 1, 2. Datasets reused in this study include Adipose expression of ATP6v0a1 in WT and leptin deficient (ob/ob) mice was obtained from expression array data (GEO GSE167264). Human adipocyte microarray expression data from lean participants and participants with obesity were downloaded from Gene expression omnibus (GEO: GSE2508)[27]. Human adipose microarray data was obtained from individuals with obesity before and after 5%, 10%, and 15% weight loss ($n = 9$ per group), GEO GSE70529[11]. All other data generated or analysed during this study are included in this published article (and its supplementary information files). Source data are provided with this paper.

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

## Acknowledgements

National Institutes of Health grant R01DK117872 (O.O. and M.P.) Larry L. Hillblom Foundation Postdoctoral Fellowship 2019-D-007-FEL (R.C.Z.) National Institute of Health grant P30 DK063491 (Diabetes Research Center, UCSD) (O.O.). "Some strains were provided by the CGC, which is funded by NIH Office of Research Infrastructure Programs (P40 OD010440)".

## Author contributions

O.O., R.C.Z., and M.P. conceived the study and designed and carried out the experiments. R.C.Z., O.O., F.C.G.R., B.S.C., J.O., D.Z., M.C., M.P., R.S, T.C. performed experiments and analysis. R.C.Z., O.O. wrote, reviewed and edited the manuscript with input from all authors.

## Competing interests

The authors declare no competing interests.
