## [Peer Review File · Nature Communications]

Title: Adipocytes control food intake and weight regain via Vacuolar-type H⁺ ATPaseEditorial Note: Parts of this Peer Review File have been redacted as indicated to remove third-party material where no permission to publish could be obtained.

REVIEWER COMMENTS

Reviewer #1 (Remarks to the Author):

Osborn and colleagues have investigated the possible role of ATP6v0a1, a subunit of V-ATPase, in adipocytes in metabolic memory. Using in vitro, nematode, and rodent models, as well as human data, they show that ATP6v0a1 is associated to changes in body mass and feeding.

This study addresses a very important question that has not been experimentally answered yet: the metabolic memory that, for example, makes regain body weight after diet. While authors propose adipose ATP6v0a1 as a possible candidate gene, some issues need clarification and required a deeper mechanistic approach.

1. Knockdown of Atp6v01a is adiponectin cre based and analyses of WAT were performed. This is a weakness, since adiponectin promoter drives cre expression in white, but also in brown and beige adipocytes. Therefore, a possible role of BAT and/or browning cannot be excluded. This is important for several reasons:

a) The % of knockdown (around 50%) seems low

b) While the main conclusion is that ATP6v0a1 ablation promotes GDF15 expression (although no mechanism is demonstrated) leading to decreased feeding, looking that the food intake graphs (4a, 6a, 7b) the anorectic effect seems not enough to explain the body weight decrease (4b, 6d, 7c). In keeping with that, the calorimetric data are not convincing; thus, the possible involvement of BAT thermogenesis and increased energy expenditure cannot be rule out. A proper analysis of EE, performing ANCOVA and correcting by body weight should be performed. Furthermore, analysis of BAT function (in normal and thermoneutral conditions) and expression markers are demanding here.

2. In relation to the former point, and the scarce magnitude of the food intake changes, paired groups are needed, as well as analysis of intestinal absorption, for example by calorimetric study of feces.

3. Hypothalamic expression data are not consistent, whereas Pomc mRNA is increased in all the ablated models, levels of AgRP and NPY show variability. Which is the reason for that?

4. The possible involvement of GDF15 is interesting but not mechanistically demonstrated. How does Atp6v01a ablation promote increased Gdf15 gene expression? Would lack of GDF15 blunt the effect of Atp6v01a knockdown on feeding and body weight?

5. In relation to the WAT changes and leptin, has the expression of Atp6v01a been addressed in ob/ob

mice?

6. The quality of western blot needs improvement. For example, Atp6v01a images in panel 3a are flawed and this is a key result of the study, as it shows the % of knockdown (which, as said, is too low). In Figure 5f some of the gels look spliced; this is OK, but if that is the case, this must be indicated with vertical lines.

Reviewer #2 (Remarks to the Author):

Zapata and colleagues present a conceptually intriguing study of metabolic memory of individuals that were obese even after losing weight. The foundation for the metabolic memory concept is strong and the study uses appropriate methodologies and models to test their hypothesis. Several statistical concerns, quality of the biochemical assessment of protein levels, and questions regarding the interpretation of the data diminish enthusiasm for the manuscript.

Major concerns:

1. The manuscript focuses on the expression levels of ATP6v0a1, but the quality of the western blots used to quantify expression levels is quite poor. Moreover, it is often difficult to assess the specificity of the Atp6v0a1 band since it does not go away in the KO? In several instances the blots are highly overexposed which makes quantification impossible.
2. Related to #1 above, while it is greatly appreciated that the authors present data for multiple replicates in their analysis, the biological replicates often do not behave the same in all lanes of analysis. This calls into question what is happening across individuals. Example: Figure 5e. pAKT and total AKT in WT and KO samples.
3. It is not clear how statistical analyses are performed. In several instances significance could be due to the a single outlier data point where in other examples sets of data that look at face very different are marked as non-significant. This is generally true across all figures, but as an example several comparisons in figure 2b between HF and switch are marked non-significant when this doesn't appear to be true.
4. The multiple models are appreciated, but it is not clear how the *C. elegans* studies relate to the main findings of the manuscript. First *C. elegans* does not have adipose tissue and it is not clear how the genes tested functionally change lipid homeostasis, which is well-studied in *C. elegans*. Do the orthologues of the memory gene in *C. elegans* also mediate metabolic memory in the worm? It seems the more relevant analysis would be to test fat worms that are made lean (similar to what was done in the mouse and human models).
5. The links to insulin and glucose handling are important and a strength of the study, but the obesity model used is a high fat diet. I am not sure what the scientific basis of this choice was and why a high carb diet wasn't employed.

Minor notes:

1. The first section of the result uses the word "primarily" which doesn't seem accurate given that SVCs are also involved.
2. The change in axis scale between figure 1d and 1e makes comparisons difficult.
3. I was unable to determine how normalization was done in multiple comparisons. What the weight gain (and loss) of each individual animal factored into the assessments?

Reviewer #3 (Remarks to the Author):

Zapata et al reported the effect of adipocyte V-ATPase on food intake and weight gain in this manuscript, they also proposed that V-ATPase is critical to establish a metabolic memory which will affect the weight gain after shifting from high fat diet to LFD. The hypothesis of metabolic memory is interesting, and the authors demonstrated retention of relatively high mRNA and protein levels of V-ATPase subunits after diet shifting in mice. However, the authors focused on characterization of the lean phenotypes in the adipocyte-specific ATP6V0a1 KO mice without providing solid data pertaining to the dependency of V-ATPase in metabolic memory. The overall quality of experimental data is low. For example, a large portion of the data on metabolic phenotypes are not solid as floxed mice were not used as a control. The major points are listed below.

1. Line 84, the subtitle "The 'metabolic memory' of obesity resides primarily in adipocytes" is not suitable, as the authors only analysed the adipose tissue, without comparison with the other tissues, such as the small intestine, skeletal muscle and hypothalamus that are also critical in regulation of whole body energy balance. Even within the adipose tissue, immune cells are not investigated either.
2. The statistical significance for Fig 2b does not echo with the individual points on the bar graph, as SW group showed apparent reduction in V0a1, V0d2, V1a and V1b2 compared to HFD group, while the graphs are labeled NS. The authors should double check their statistical methods.
3. Fig2e, actin is overexposed (hollow) in the representative blot, how can the authors derive meaningful densitometry data from such data?
4. For human samples, the elevation in the mRNA of V-ATPase subunits in obese people is very mild, what about the protein levels?
5. ANOVA analysis using body weight as a covariate should be used when analyzing food intake and energy expenditure. Current conclusions on these data in ATP6V0a1 KO mice need to be verified. The authors report in their method section that they used lean mass +20% body fat as a covariate in energy expenditure data; however, the related data is not presented as X-Y graphs. Besides, total body weight should be used as covariate in such analysis (specified in ref 73).
6. The authors only used Cre mice as a control for a large portion of the data (such as Fig 4 c-k), while it is obviously not appropriate to exclude F/F mice data, especially the protein levels of ATP6V0a1 is not unambiguously depleted in the F/F-cre mice.
7. It is not clear whether the basal plasma glucose levels as well as insulin levels are changed in KO mice under fed or fasting states.
8. Skeletal muscle which has a similar role in controlling circulating glucose in response to insulin should

also be used to demonstrate the tissue specificity of the insulin response. The authors show that the insulin sensitivity in the liver is not changed in KO mice. Is this because the variation among individuals is too high?

9. Bafilomycin has a wide spectrum of intracellular targets, the effects of which cannot be interpreted as V-ATPase inhibition. Importantly, it seems that the authors did not perform experiments using KO and control mice in the same experiments, as the graphs showing BFM effects are separately demonstrated as WT and KO (Fig6).

10. There is no data directly showing the effect of ATP6v0a1 on weight regain, which will be a fundamental piece of data to support the metabolic memory hypothesis. The authors need to include KO mice or use other strategies that can target ATP6v0a1 more specifically in such analysis.

11. Fig 8, the mechanism for ATP6v0a1 in weight regain is elusive. The effect of adipocytes on whole body energy balance could be attributed to local lipid storage defects or adipokine-mediated organ crosstalk. The authors claim GDF15 is upregulated in response to ATP6v0a1 downregulation in adipocytes; however, secretion of the other well-known adipokines such as leptin or adiponectin are not investigated in the cell model. In fact, even the upregulation of GDF15 is very mild.

12. Primary adipocytes from mice fed with NCD, HFD or SW need to be used to demonstrate the metabolic memory effect. 3T3-L1 model in Fig 8 cannot support metabolic memory hypothesis.

We wish to thank the reviewers for their insightful comments and questions and we provide a point by point response to each below.

Reviewer#1

This study addresses a very important question that has not been experimentally answered yet: the metabolic memory that, for example, makes regain body weight after diet. While authors propose adipose *Atp6v0a1* as a possible candidate gene, some issues need clarification and required a deeper mechanistic approach.

1. Knockdown of *Atp6v01a* is adiponectin cre based and analyses of WAT were performed. This is a weakness, since adiponectin promoter drives cre expression in white, but also in brown and beige adipocytes. Therefore, a possible role of BAT and/or browning cannot be excluded. This is important for several reasons:
 A). The % of knockdown (around 50%) seems low.

We have now conducted comprehensive expression analysis of *Atp6v0a1* expression in white (WAT) and brown adipose tissue (BAT) as well as fractionated adipocytes. At the RNA level, *Atp6v0a1* expression in the KO is decreased approximately 73% in WAT and relatively unchanged in BAT (16% reduction) compared with WT controls. We also measured expression of genes related to BAT function and determined no significant differences in *Ucp1*, *Cidea*, *Pgc1a*, *Prdm16*, *Cyt3* or *Dio2* (Supplemental Fig 2d) further confirming that this small decrease in *Atp6v0a1* expression did not impact key BAT function genes.

We have also repeated the fractionation of adipocytes from white adipose tissue in WT and KO mice and found the protein level of *Atp6v0a1* was 92% lower in adipocytes from KO mice compared with WT, while no differences in *Atp6v0a1* expression were observed in the stromal vascular cells (Fig 3b).

Figure 3

Supplemental Fig 2

B) While the main conclusion is that ATP6v0a1 ablation promotes GDF15 expression (although no mechanism is demonstrated) leading to decreased feeding, looking that the food intake graphs (4a, 6a, 7b) the anorectic effect seems not enough to explain the body weight decrease (4b, 6d, 7c).

This is an important point and further discussion related to this has now been added to the manuscript. While the daily average differences in food intake may appear small, these differences in daily food intake can have a highly significant cumulative effect over time. We summarize examples of how food intake differences relate to changes in body weight in each of our experiments below:

Figure 3c-d. In chow fed mice, KO ate 10% less compared with Cre (WT) or FF mice and weigh 13% less than Cre (WT) and 8% less than FF mice.

Figure 4a-b. KO phenotype on HFD (eat ~11% less compared with WT, or 23% compared with Flox mice) leading to a 20% less weight gain compared with WT or 25% less than FF.

Figure 6a-b. HFD fed bafilomycin treated mice ate 10% less and weight 6% less than vehicle treated.

Figure 7b-c. HFD fed bafilomycin treated “weight regain” mice ate 15% less food, weighed 9% less (regained 4.65 g less) over the 21 day treatment.

Therefore, in all of our experiments, lower food intake consistently leads to lower body weight, but the magnitude of the effect depends on the diet administered and the time course studied. For example, chronically fed chow fed KO mice eat 10% less and have an 8-13% lower body weight than WT mice. In a similar model where C57Bl6 male mice were chronically calorie restricted (20% less calories for 9 months), this resulted in 14% lower body weight compared with *ad libitum* fed control mice (PMID 27304509). In contrast, upon switching mice to high fat diet, the differences in food intake (g) have a significantly greater effect on weight gain with an average of 11% less food intake leading to 20-25% lower weight gain due to the high caloric density of the high fat diet. Therefore, diet and duration of feeding are important parameters that contribute significantly to overall body weight. Adaptive processes can also counteract acute changes in diet over time. In summary, the observed changes in food intake in our studies are closely in line with expected and observed changes in body weight.

c) In keeping with that, the calorimetric data are not convincing; thus, the possible involvement of BAT thermogenesis and increased energy expenditure cannot be ruled out. A proper analysis of EE, performing ANCOVA and correcting by body weight should be performed.

We thank the reviewers for this important point and have now re-analyzed the metabolic chamber data and performed ANCOVA correcting for body weight (PMID 31391589). Interestingly, this analysis revealed that the KO mice actually have a slightly lower metabolic rate than the WT and this difference is particularly apparent during the light cycle when mice are resting (**Fig 4e**, and **Supplemental Fig 2 a-b**). This further confirms that the lower body weight observed in the KO is driven by lower food intake, and not increased energy expenditure.

Supplemental figure 2

Figure 4e

d) Furthermore, analysis of BAT function (in normal and thermoneutral conditions) and expression markers are demanding here.

To address this point, we have now measured expression of BAT function genes (Ucp1, Cidea, Pgc1a, Prdm16, Cyt3 or Dio2) in WT and KO mice and found no significant differences in expression of in BAT (**see supplemental figure 2d**). Furthermore, expression levels of Atp6v0a1 were unchanged in BAT from mice challenged with acute cold exposure for 2-4 days (PMID 25516548, Supplemental Table 2). Similarly, RNA sequencing analysis of beige (inguinal)

adipose tissue from mice exposed chronically to cold temperature revealed no changes in expression of *Atp6v0a1* (PMID 25516548, 34017267). Adipose tissue *Atp6v0a1* expression was also unchanged after pharmacological induction of beigeing using a beta-3 adrenergic agonist (CL-316,243) (PMID 34017267, Supplemental Material 2). Taken together, these data suggest *Atp6v0a1* ablation does not have an overt impact on BAT function.

Supplemental Figure 2

2. In relation to the former point, and the scarce magnitude of the food intake changes, pair-fed groups are needed, as well as analysis of intestinal absorption, for example by calorimetric study of feces.

We thank the reviewers for this suggestion and we believe this pair-feeding data has greatly enhanced our study. WT mice were pair fed the same amount calories the KO mice consume. As anticipated, before the start of the experiment WT mice were heavier than KO mice. The WT mice were then divided into *ad libitum* and pair fed groups. After the 7-days, the WT pair-fed mice weighed significantly less than the WT *ad libitum* group and had similar body weights as the KO group (**Supplemental Fig 2e-f**). This strongly suggests that the lower body weight observed in the KO mice is due to lower caloric intake and not due to differences in intestinal absorption.

Supplemental Fig 2e-f

3. Hypothalamic expression data are not consistent, whereas *Pomc* mRNA is increased in all the ablated models, levels of *AgRP* and *NPY* show variability. Which is the reason for that?

As noted by the reviewer, we observe a consistent increased in anorexigenic neuropeptide *Pomc* in all comparisons (**Fig 4h**= KO v WT, **Fig 6i**= Bafilomycin treatment, **Fig 7e**=Bafilomycin treatment in weight rebound). However, the levels of orexigenic *Npy* were significantly decreased

in KO v WT comparisons (**Fig. 4h**) but unchanged by bafilomycin treatment (**Fig 6i** and **7e**). Furthermore, bafilomycin significantly reduced *Agrp* expression in WT mice (**Fig 6i**) but levels were not significantly different in HFD treated weight rebound mice (**Fig 7e**).

There are multiple explanations for this variability in neuropeptide expression in response to ATP6v0a1 deletion or inhibition. Expression levels of neuropeptides can vary due to recent food intake, stress and circadian rhythms (PMID 28373831). To minimize the impact of these effects we consistently sacrificed the mice during the light period to diminish the effect of circadian rhythms. However, mice were sacrificed under normal fed states and thus potentially some mice may have consumed food more recently than others resulting in acute changes in neuropeptide expression.

Fig 4

Fig 6

Fig 7

4. The possible involvement of GDF15 is interesting but not mechanistically demonstrated. How does *Atp6v01a* ablation promote increased *Gdf15* gene expression?

We agree with the reviewer that the link between ATP6v0a1 and Gdf-15 is particularly interesting. ATP6v0a1 is a key component of the V-ATPase that acts as a proton pump that generates an acidic environment in the lysosome. If lysosomal activity is perturbed, this leads to the activation of Ddit3 (Chop) signaling pathway and results in ER Stress. Recent literature has highlighted a mechanistic link between ER stress and Gdf-15 levels (PMID28847729, 33302552). Notably, Chop binds to the promoter of GDF15 and activates its transcription under ER stress conditions (PMID 28847729, PMID33302552). Therefore, to evaluate whether ATP6V0a1 triggered ER stress, we measured the expression of Ddit3 and activating transcription factor 4 (ATF4), which are classical markers of ER stress, in ATP6 KO and Bafilomycin treated WT mice compared with controls.

Ddit3 and *Atf4* were significantly upregulated in the EWAT of KO mice compared with WT (**Fig. 4i**). Furthermore, bafilomycin treatment also upregulated expression of both Ddit3 and *Atf4* (**Fig. 7g**). We also treated 3T3L1 adipocytes with ATP6v0a1 siRNA and observed a significant increase in ER stress markers (**Fig. 8e**). Therefore, inhibition of lysosomal activity through genetic and pharmacological manipulations of V-ATPase/*Atp6v0a1* result in increased ER stress. The transcription factor Ddit3 (Chop) binds directly to the *Gdf-15* promoter region and regulates its expression (PMID 28847729, PMID 33302552). This provides a potential mechanism through which ATP6v0a1 ablation or pharmacological inhibition increases expression of Ddit3 (Chop) and drives increased *Gdf-15* expression.

Fig 4**Fig 7****Fig 8**
5. In relation to the WAT changes and leptin, has the expression of Atp6v01a been addressed in ob/ob mice?

The expression of ATP6v0a1 is highly elevated in multiple models of obesity including high fat diet fed mice and leptin deficient (ob/ob) mice fed either normal chow or high fat diet (RNA seq data downloaded from GEO GSE 167264). This confirms that obesity, whether induced by high fat diet, or leptin deficiency drives significant elevation in expression of ATP6v0a1. (See **Figure 2c**)

6. The quality of western blot needs improvement. For example, Atp6v01a images in panel 3a are flawed and this is a key result of the study, as it shows the % of knockdown (which, as said, is too low). In Figure 5f some of the gels look spliced; this is OK, but if that is the case, this must be indicated with vertical lines.

We agree with the reviewer and have repeated these experiments and further optimized conditions to improve these blots. We re-isolated adipocytes from the adipose tissue of WT and KO mice and we now show, with clearer blots, that KO decreased Atp6v0a1 protein level by ~80%. In addition, we have repeated the western blot of the liver of WT and KO mice that were acutely stimulated with insulin, we isolated protein samples again and we now show, with cleaner blots, that there are no significant effects of Akt phosphorylation between WT and KO mice.

Figure 3b**Figure 5**
Reviewer #2

Zapata and colleagues present a conceptually intriguing study of metabolic memory of individuals that were obese even after losing weight. The foundation for the metabolic memory concept is strong and the study uses appropriate methodologies and models to test their hypothesis. Several statistical concerns, quality of the biochemical assessment of protein levels, and questions regarding the interpretation of the data diminish enthusiasm for the manuscript.

Major concerns:

1. The manuscript focuses on the expression levels of ATP6v0a1, but the quality of the western blots used to quantify expression levels is quite poor. Moreover, it is often difficult to assess the specificity of the Atp6v0a1 band since it does not go away in the KO? In several instances the blots are highly overexposed which makes quantification impossible.

We have repeated many of the western blots in this manuscript, further optimized conditions and significantly improved their quality. We note a highly significant reduction in Atp6v0a1 expression in the KO (92%, Figure 3b) and further confirmed antibody specificity by using siRNAs to deplete expression of ATP6v0a1 in 3T3 cells (Figure 8b).

Figure 3b

Figure 8b

2. Related to #1 above, while it is greatly appreciated that the authors present data for multiple replicates in their analysis, the biological replicates often do not behave the same in all lanes of analysis. This calls into question what is happening across individuals. Example: Figure 5e. pAKT and total AKT in WT and KO samples.

We have re-isolated protein samples and further optimized western blot conditions for this experiment. We now show that knockout of ATP6v0a1 in adipocytes results in significant enhancement of insulin sensitivity in adipose tissue (elevation in insulin-stimulated pAkt). In addition, adipocyte KO of Atp6v0a1 has no impact on liver or muscle insulin sensitivity in liver or skeletal muscle. See new **Figure 5**.

Figure 5

3. It is not clear how statistical analyses are performed. In several instances significance could be due to a single outlier data point where in other examples sets of data that look at face very different are marked as non-significant. This is generally true across all figures, but as an example several comparisons in figure 2b between HF and switch are marked non-significant when this doesn't appear to be true.

Figure 2b

We have checked all stats and added further details to the figure legends. **Figure 2b** is visualization of a subset of data from our RNA-seq study. All stats are derived from the full analysis of all genes and not just this subset presented here. While there is a trend in lower expression of some of these subunits after diet switch, these changes in expression are not statistically significant when corrected for the number of comparisons made. Additional details related to the statistics used have been added to the figure legend. Biostatistician Roman Sasik (author) played a key role in the data analysis of these studies.

4. The multiple models are appreciated, but it is not clear how the *C. elegans* studies relate to the main findings of the manuscript. First *C. elegans* does not have adipose tissue and it is not clear how the genes tested functionally change lipid homeostasis, which is well-studied in *C. elegans*. Do the orthologues of the memory gene in *C. elegans* also mediate metabolic memory in the worm? It seems the more relevant analysis would be to test fat worms that are made lean (similar to what was done in the mouse and human models).

Model organisms provide an innovative way to conduct phenotypical screens of read outs that are beyond the capabilities of cell lines. In this assay, *C. elegans* act like a cell line that 'eats' which facilitates the opportunity to use worms to study food intake in a robust and reproducible assay. These studies rely on identification of highly homologous genes between worm and mouse and identifying their potential role in food intake. We have used a similar phenotypic screen to identify modulators of food intake in a recent study related to overfeeding (PMID 30532051). However, while *C. elegans* allows highly precise quantification of food intake, worms do not accumulate adipose tissue in the same way as higher mammals. Importantly, in our assay, worms consume bacteria, which is not rich in fat or sugar per se. Nevertheless, feeding worms higher concentrations of fructose, sucrose and glucose significantly shortened the nematode's maximum length. Similarly, adding stearic acid, cholesterol and linoleic acid, in general, also do not affect growth (PMID 32015763). We have utilized this innovative *C. elegans* phenotypic screening strategy to identify genes implicated in food intake and conducted follow up studies in mice where adipose tissue biology is much better understood and with greater translational relevance.

5. The links to insulin and glucose handling are important and a strength of the study, but the obesity model used is a high fat diet. I am not sure what the scientific basis of this choice was and why a high carb diet wasn't employed.

The 60% HFD diet is the most commonly used diet in mouse obesity research as it accelerates weight gain and results in significant increase in fat mass. However, recent discussion on the topic of high fat diet composition in relation to the human diet highlight the importance of consideration of macronutrient content in the context of mimicking human dietary habits (PMID 30967607). Our data shows ATP6v0a1 is also elevated in genetic models of obesity (Fig 2c) and adipocytes from human obese individuals (Fig 2e) suggesting the effects are independent of diet. This important point is now added to the discussion.

Minor notes:

1. The first section of the result uses the work "primarily" which doesn't seem accurate given that SVCs are also involved.-

We have now amended this title to "Adipocytes retain the "metabolic memory" of obesity"

2. The change in axis scale between figure 1d and 1e makes comparisons difficult.

We have adjusted the scales in **Fig 1e** to be the same **Fig 1d**. The scale in Fig 1d is food intake in a basal state, while Fig 1e is showing the response of the WT (N2) and mutant strain (*unc-32*) in response to the hyperphagic stimulus (serotonin) where food intake is induced to a higher level up to (333%).

Figure 1

3. I was unable to determine how normalization was done in multiple comparisons. What the weight gain (and loss) of each individual animal factored into the assessments?

In the *C. elegans* assay, food intake was normalized to N2 ('WT') worms. This is described in the Methods section (lines 526-528). In the mouse studies, food intake data are presented as grams consumed, without any normalization.

Reviewer #3

Zapata et al reported the effect of adipocyte V-ATPase on food intake and weight gain in this manuscript, they also proposed that V-ATPase is critical to establish a metabolic memory which will affect the weight gain after shifting from high fat diet to LFD. The hypothesis of metabolic memory is interesting, and the authors demonstrated retention of relatively high mRNA and protein levels of V-ATPase subunits after diet shifting in mice. However, the authors focused on characterization of the lean phenotypes in the adipocyte-specific ATP6v0a1 KO mice without providing solid data pertaining to the dependency of V-ATPase in metabolic memory. The overall quality of experimental data is low. For example, a large portion of the data on metabolic phenotypes are not solid as floxed mice were not used as a control. The major points are listed below.

1. Line 84, the subtitle the "metabolic memory" of obesity resides primarily in adipocytes is not suitable, as the authors only analyzed the adipose tissue, without comparison with the other tissues, such as the small intestine, skeletal muscle and hypothalamus that are also critical in regulation of whole body energy balance. Even within the adipose tissue, immune cells are not investigated either.

In previous studies, we have investigated the 'memory' of key metabolic tissues (adipose, liver, hypothalamus, skeletal muscle and plasma) after weight loss (PMID 29089614). In our lipidomics based experiment, we found the greatest number of persistently changed lipid mediators are found in the adipose tissue (PMID 29089614) compared with the other tissues after weight loss. We have also conducted RNA seq on muscle tissues after weight loss and found the majority of gene expression changes reverted back to pre-obese levels after weight loss (PMID 29634311). In addition, we have studied transcriptional changes in the hypothalamus and liver after high fat

diet feeding and weight loss (not yet published), and these findings corroborate the findings from our lipidomics based studies (PMID 29089614) that the majority of the metabolic memory of obesity resides in adipose tissue. Notably, the adipose tissue plays an important role in the regulation of energy balance and has been the focus of many theories aimed at explaining the persistence of a biological drive to regain weight (PMID 31742247, 29089614, 25614203, 27110485, 30655624). Adipose tissue is a heterogenous cellular population, so we fractionated it into both adipocytes and the immune cell contain fraction (stromal vascular cells, SVCs) and conducted RNA seq on both subsets of cells in lean and obese mice. In a comparison of lean vs obese mice, we found 925 adipocyte genes were differentially expressed compared with 354 genes in the SVCs. Furthermore, 752/925 adipocyte genes remained persistently dysregulated after weight loss, while 247/354 were persistently dysregulated in SVCs. We have now modified our discussion to reflect the fact that there are also persistently dysregulated genes in the immune cell fraction that may play also an important role in metabolic memory.

2. The statistical significance for Fig 2b does not echo with the individual points on the bar graph, as SW group showed apparent reduction in V0a1, V0d2, V1a and V1b2 compared to HFD group, while the graphs are labeled NS. The authors should double check their statistical methods.

We have checked all stats and added further details to the figure legends. Figure 2b is a visualization of a subset of data from our comprehensive RNA-seq study. All stats are derived from the full analysis of all samples and not just this subset presented here. While there is a trend in lower expression of some of these subunits after diet switch, these changes in expression are not statistically significant when corrected for the number of comparisons made. Additional details have been added to the figure legend.

Figure 2b

3. Fig2e, actin is overexposed (hollow) in the representative blot, how can the authors derive meaningful densitometry data from such data?

We have now re-run this blot and present new Gapdh data in **Fig 2g-h**. Quantification of Gapdh on the same samples confirms the findings that abundance of ATP6v0a1 increases in HFD fed compared with lean in both membrane bound and cytosolic forms. In addition, levels of ATP6v0a1 do not return to pre-obese levels after weight loss (SW= formerly obese)

Figure 2g-h

4. For human samples, the elevation in the mRNA of V-ATPase subunits in obese people is very mild, what about the protein levels?

We observed elevated gene expression levels of ATP6v0A1 in human visceral adipose tissue in obese subjects compared with lean individuals (Fig 2d). In addition, we have now studied ATP6v0a1 levels in adipocytes from subcutaneous adipose tissue from lean, (BMI of 25.15; range 23.5-26.5) and obese (BMI of 35.1; range 33.1-38.2 (non-diabetic) humans, consistent with transcript levels. While protein levels of ATPv0a1 were variable between individuals, levels were significantly elevated in obese subjects compared to lean subjects, as presented in Fig 2e, and these levels do not decrease after weight loss (Fig 2f).

Figure 2

5. ANOVA analysis using body weight as a covariate should be used when analyzing food intake and energy expenditure. Current conclusions on these data in ATP6V0a1 KO mice need to be verified. The authors report in their method section that they used lean mass +20% body fat as a covariate in energy expenditure data; however, the related data is not presented as X-Y graphs.

We thank the reviewers for this important point and have now re-analyzed the metabolic chamber data and performed ANCOVA correcting for body weight (PMID 31391589). Interestingly, this analysis revealed that the KO mice have a slightly lower metabolic rate than the WT and this difference is particularly apparent in during the light cycle, when mice are resting (Fig 4e, and

Supplemental Fig 2 a-b). This further confirms that the lower body weight observed in the KO is driven by lower food intake, and not increased energy expenditure.

We have also re-analyzed the food intake using ANCOVA with body weight as co-variate. Similar results were found compared to the previous analysis wherein KO mice have reduced food intake compared to Control (cre) and Floxed (F/F) (**supplemental fig 2c**)

Suppl Fig 2

Fig 4e

6. The authors only used Cre mice as a control for a large portion of the data (such as Fig 4 c-k), while it is obviously not appropriate to exclude F/F mice data, especially the protein levels of ATP6V0a1 are not unambiguously depleted in the F/F-cre mice.

Importantly, we have confirmed that Cre and FF mice have similar levels of ATP6v0a1 and a similar metabolic phenotype. Both F/F and Cre mice have similar food intake (**Fig. 3c, Fig. 4a**) and body weight (**Fig. 3d, Fig. 4b**). We also measured the expression of Atp6v0a1 at the both the gene and protein level in the adipose tissue and determined that Cre and F/F mice have equivalent ATP6v0a1 levels (**Supplemental Fig 1a-b**)

Figure 3

Figure 4

Supplemental Fig 1a-b

7. It is not clear whether the basal plasma glucose levels as well as insulin levels are changed in KO mice under fed or fasting states.

The fasted basal glucose levels were slightly lower in the HFD-fed KO compared with WT (**Fig 5a**; KO: 178 mg/dL, WT: 202mg/dL, $p=0.15$). Fasted insulin also followed a similar trend whereby KO mice have lower levels of fasted insulin (**Fig 5c**; KO: 1.64, WT: 2.07 ng/ml, $p=0.17$).

We have also measured Fed/non-fasting glucose levels which were also not significantly different between WT and KO groups, (shown below, not included in manuscript)

8. Skeletal muscle which has a similar role in controlling circulating glucose in response to insulin should also be used to demonstrate the tissue specificity of the insulin response. The authors show that the insulin sensitivity in the liver is not changed in KO mice. Is this because the variation among individuals is too high?

We have conducted comprehensive analysis of basal and insulin-stimulated pAkt in WAT, liver and skeletal muscle (**Fig 5f-i**). Male 8-week HFD-fed KO and WT mice were anesthetized and basal liver, muscle and WAT biopsies were obtained before administration of 0.7 U/kg insulin through the portal vein. Insulin stimulated tissue biopsies of the liver, muscle and WAT were then collected at 3, 5 and 7 minutes, respectively, and then frozen at -80°C and later processed to measure pAKT relative to total AKT. These studies revealed that KO mice have increased WAT insulin sensitivity, while no significant differences in insulin stimulated pAKT were observed in liver or skeletal muscle in KO compared with WT mice.

9. Bafilomycin has a wide spectrum of intracellular targets, the effects of which cannot be interpreted as V-ATPase inhibition. Importantly, it seems that the authors did not perform experiments using KO and control mice in the same experiments, as the graphs showing BFM effects are separately demonstrated as WT and KO (Fig 6).

These experiments were conducted at the same time but were previously presented independently for clarity. We have now modified the figures accordingly. We have left the GTT and ITT as separate graphs as with all 4 groups the lines overlap and it is hard to visualize. (**see new Figure 6**).

Figure 6

10. There is no data directly showing the effect of ATP6v0a1 on weight regain, which will be a fundamental piece of data to support the metabolic memory hypothesis. The authors need to include KO mice or use other strategies that can target ATP6v0a1 more specifically in such analysis.

Models of spontaneous weight regain with high translational relevance are challenging in mice. This is mainly due to the fact mice have a preference to consume HFD over normal chow and thus when exposed to HFD, they gain weight due to the high caloric density compared with normal chow food. This model lacks many elements of food intake behavior (including food choice, food reward etc) that play a key role in human weight re-gain. However, some mouse-based studies have observed subtle differences in the rate of weight gain in mice exposed to HFD a second time (PMID 27906159) . Therefore, we conducted this weight re-gain assay in BFM treated mice. The differences in rate of weight gain between first and second exposure of HFD were not statistically significant (WT vs WT-Rb-VEH; **see graph below**), suggesting the subtle observations of rate of weight regain are not highly reproducible. However, importantly, we note that BFM treatment reduces weight regain in formerly obese mice compared with VEH. We have also shown that the action of BFM to reduce food intake and weight gain is dependent on ATP6v0a1 expression in adipose tissue (**Fig 6a and 6b**). Moreover, when adipocyte specific ATP6v0a1 KO were treated with BFM there were no significant differences in food intake or weight gain.

11. Fig 8, the mechanism for ATP6v0a1 in weight regain is elusive. The effect of adipocytes on whole body energy balance could be attributed to local lipid storage defects or adipokine-mediated organ crosstalk. The authors claim GDF15 is upregulated in response to ATP6v0a1 downregulation in adipocytes; however, secretion of the *other well-known adipokines such as leptin or adiponectin* are not investigated in the cell model. In fact, even the upregulation of GDF15 is very mild.

We measured expression of leptin, adiponectin and gdf-15 in adipocytes with specific knockdown of Atp6v0a1. Atp6v0a1 siRNA treatment of adipocytes resulted in 54% knockdown at mRNA level (Fig 8a) and 33% lower protein levels (Fig 8b) that was associated with two fold elevation in appetite suppressing hormone Gdf-15 mRNA (Fig 8c) and 20% increase in Gdf-15 protein levels (Fig 8d).

Leptin and adiponectin mRNA levels were unchanged by Atp6v0a1 siRNA treatment (Fig 8c) while lower leptin levels were observed in KO mice (Fig 4i) and BFM treated WT mice (Fig 7g) which likely reflect the lower fat mass in these mice.

ATP6v0a1 is a key component of the V-ATPase that acts a proton pump that generates an acid environment in the lysosome. When normal lysosomal activity is perturbed this leads to ER stress which includes activation of the Ddit3 (Chop) signaling pathway. Recent literature has revealed that activation of ER stress also triggers increased levels of appetite suppressing hormone Gdf-15 (PMID 28847729, PMID 33302552). These studies determined that Ddit3 binds directly to the promoter of GDF15 and activates its transcription under ER stress conditions (PMID 28847729, 33302552). Therefore, to evaluate whether Atp6V0a1 depletion triggered ER stress we measured the expression of key genes implicated in ER stress, including Ddit3 and activating transcription factor 4 (ATF4). Atp6v0a1 knockdown was associated with significantly increased levels of Ddit3 (22%) and Atf4 (21%) (Fig. 8c). In addition, we observed significantly upregulated levels of *Chop* and *Atf4* in eWAT from both ATP6v0a1 KO mice (Fig 4i) and BFM treated WT mice (Fig 7g), compared with WT controls.

Therefore, inhibition of lysosomal activity through genetic and pharmacological manipulation of V-ATPase/Atp6v0a1 results in increased activation of ER stress pathways. Recent studies have revealed that transcription factor Ddit3 binds directly to Gdf-15 promoter region and regulates its expression. This provides a potential mechanism through which ATP6v0a1 ablation or pharmacological inhibition increases expression of Ddit3 and drives increased Gdf-15 expression. This important new data is now added to the manuscript and included in the discussion.

Fig 4i**Fig 7g****Figure 8**
12. Primary adipocytes from mice fed with NCD, HFD or SW need to be used to demonstrate the metabolic memory effect. 3T3-L1 model in Fig 8 cannot support metabolic memory hypothesis.

Our studies show that Atp6v0a1 expression is elevated in adipocytes from obese mice and humans, and that this elevation persists despite weight loss. Our mouse-based adipocyte specific KO studies revealed that Atp6v0a1 plays an important role in food intake and body weight regulation. We also show that pharmacological inhibition blunts weight regain formerly obese mice that are challenged with HFD. We have also cultured 3T3L1 adipocytes and specifically knocked down Atp6v0a1 expression that resulted in elevation of ER stress genes including Ddit3 and downstream elevation in satiety promoting hormone Gdf15. These 3T3L1 cellular based experiments provide a potential mechanism underlying ATP6v0a1 mediated regulation of food intake and body weight. However, our in vivo studies provide the strongest and most compelling evidence for the role of At06v0a1 in metabolic memory.

REVIEWERS' COMMENTS

Reviewer #1 (Remarks to the Author):

All my comments have been addressed.

Reviewer #2 (Remarks to the Author):

The authors have addressed my concerns. This is an excellent study with interest to the broad readership of the journal.

Reviewer #3 (Remarks to the Author):

All my concerns have been addressed in this revised manuscript. I would like to recommend publication of this paper.